# Single-event fast neutron time-of-flight spectrometry with a petawatt-laser-driven neutron source

M. A. Millán-Callado [1,2,12,13], S. Scheuren [3,13], A. Alejo [4], J. Benlliure [4,5], R. Beyer [6], T. E. Cowan [6,7], B. Fernández [1,2], E. Griesmayer [8,9], A. R. Junghans [6], J. Kohl[3], F. Kroll [6], J. Metzkes-Ng [6], I. Prencipe[6], J. M. Quesada [2], M. Rehwald [6], C. Rödel [3,10], T. Rodríguez-González [1,2], U. Schramm [6,7], M. Roth[3,11], R. Štefaníková [6,7], S. Urlass[6], C. Weiss [8,9], K. Zeil [6] ✉, T. Ziegler [6] & C. Guerrero [1,2] ✉

Laser-driven neutron sources (LDNSs) offer unique advantages for fundamental physics and applications: ultrashort pulses providing superior energy resolution, high instantaneous flux, and a reduced footprint. While single-event neutron spectroscopy has been demonstrated with epithermal neutrons, its application for fast neutrons is more challenging and remains unproven. This demands stable multi-shot operation and detectors resilient to this particularly extreme environment. Here, a proof-of-concept experiment at the DRACO PW laser is presented. This setup stably produced ~$10^8$ fast neutrons per shot sustained over more than 200 shots at a shot-per-minute rate. Neutron time-of-flight measurements with a diamond detector at only 150 cm from the source resolved individual neutron-induced reactions at a rate consistent with simulations informed by real-time diagnostics of accompanying gammas, ions, and electrons. Combined with the recent advances in the field, this work establishes LDNSs as a promising, scalable platform for future fast neutron-induced reaction studies, particularly those involving short-lived isotopes.

Neutron-induced reactions play a crucial role in various research fields, from exploring the origins of elements in our universe[1] to understanding the core principles of nuclear physics[2,3]. These reactions are also relevant to dosimetry[4], medicine[5,6], space science[7], material science[8], condensed matter physics[9], and nuclear fission and fusion technology[10], highlighting their broad scientific, societal, and industrial relevance.

As many research reactors are decommissioned, particle accelerators take the lead in experimental neutron science. To study nuclear structures, fission dynamics, material composition, and nucleosynthesis processes at high resolution and across a broad neutron energy range, accelerator-based neutron sources commonly employ the neutron time-of-flight (ToF) method. Pulsed neutron beams are

[1]Centro Nacional de Aceleradores (CNA), US-Junta de Andalucía-CSIC, Seville, Spain. [2]Dpt. Física Atómica, Molecular y Nuclear (FAMN), Facultad de Física. Universidad de Sevilla (US), Seville, Spain. [3]Technical University of Darmstadt, Institute for Nuclear Physics, Darmstadt, Germany. [4]Instituto Galego de Física de Altas Enerxías (IGFAE), Universidad de Santiago de Compostela, Santiago de Compostela, Spain. [5]Instituto de Física Corpuscular (IFIC), CSIC - Universitat de València, Valencia, Spain. [6]Helmholtz-Zentrum Dresden – Rossendorf (HZDR), Dresden, Germany. [7]TUD Dresden University of Technology, Dresden, Germany. [8]CIVIDEC Instrumentation GmbH, Vienna, Austria. [9]Technische Universität Wien, Vienna, Austria. [10]Schmalkalden University of Applied Sciences, Schmalkalden, Germany. [11]Focused Energy GmbH, Darmstadt, Germany. [12]Present address: Currently Physikalisch-Technische Bundesanstalt (PTB), Braunschweig, Germany. [13]These authors contributed equally: M. A. Millán-Callado, S. Scheuren. ✉e-mail: k.zeil@hzdr.de; cguerrero4@us.es

generated and travel a well-defined flight path to a sample, where they induce nuclear reactions. To measure, e.g., neutron-energy-dependent reaction cross-sections, individual reactions can be detected via promptly emitted radiation. This enables the determination of the flight time—and thus the kinetic energy—of the interacting neutron, with the energy resolution primarily governed by the initial neutron pulse duration and the flight path length.

Nuclear reactions induced by fast neutrons (>1 MeV) are of particular importance for fission, fusion, and dosimetry, as reflected in the Nuclear Energy Agency's High Priority Request List, where they account for over 90% of the requested nuclear data[11]. Additionally, more than 25% of requests concern radioactive actinide isotopes. The same applies to neutron capture in astrophysics[12], in which the s-process branching points of interest are all radioactive. Here, the pulse intensity is particularly advantageous for measurements on said radioactive samples as it improves the ratio between signal and radioactive background[13]. As a result, there is increasing interest in exploring alternative sources that can deliver intense, energetic neutron pulses to address these experimental needs.

Even though the world's largest ion accelerators host proton spallation facilities that provide such neutron pulses at ~Hz repetition rates[14–18], the nuclear physics community is looking to expand the range of neutron sources to meet current and future demands. In this context, advanced compact ion sources based on laser-plasma acceleration (LPA)—a rapidly advancing technology—have been discussed as drivers of fast neutron sources to complement and extend the existing infrastructure for nuclear physics research using the neutron time-of-flight method[19].

In LPA, an ultra-short (fs to ps) high-power (TW to PW) laser pulse interacts with a solid target, generating ultra-short (~ps) ion pulses in a plasma process accompanied by a strong prompt emission of electrons, X-rays, and electromagnetic noise. The acceleration process favors ions with a high charge-to-mass ratio, in particular protons. LPA proton pulses contain up to $10^{12}$ protons at MeV energies and exhibit a broad angular distribution and an exponentially decaying energy spectrum with a cut-off towards higher kinetic energies. It has recently been demonstrated that petawatt-class (PW) lasers operating at Hz-level repetition rates can reliably produce intense proton pulses up to several tens of MeV[20,21].

Owing to the characteristics of LPA beams, the corresponding laser-driven neutron sources (LDNSs) offer an ultra-high instantaneous flux (e.g., $10^{11}$ neutrons/shot[22]), which presents a dual advantage for nuclear reaction studies with neutron time-of-flight: First, for a given energy resolution, their short pulse duration allows for a short flight path length and therefore compact setup dimensions. Second, as discussed above, a high instantaneous neutron count is particularly advantageous for measurements on radioactive samples.

LDNSs have been experimentally realized and investigated, most commonly by directing LPA protons or deuterons onto neutron converter targets in various configurations[23–25] (see Ref. 26 for the most recent review). Proof-of-concept experiments have shown the use of laser-driven neutrons for radiographic imaging[27], material analysis[28,29], and as diagnostic tools in laser-plasma physics[30–32]. Among these, the most studied application compatible with current LDNS capabilities is neutron resonance spectroscopy. This is a non-destructive material analysis technique where the dips in a transmission measurement are unambiguously assigned to a given isotope. The technique has been achieved both in a single shot, operating a neutron detector in current mode[33], and by averaging over a few shots, employing single-neutron detection[22,34]. However, to date, experimental work has been limited to moderated sources delivering epithermal neutrons in the eV range. For unmoderated fast-neutron sources, only prospective studies on resonance spectroscopy or single-neutron detection have been reported[29]. Consequently, more complex measurements approaching full nuclear-

physics experiments remain unexplored. Even if they have been recognized as a key scientific case at upcoming world-class laser facilities like ELI-NP[35], their feasibility has yet to be experimentally demonstrated.

To show that a future research program on neutron-induced nuclear reactions with LDNSs is viable, three key capabilities must be established:

(1) It is necessary to obtain as complete information as possible about individual reactions and interaction events (i.e., single-event detection). This high degree of fidelity requires a measurement technique that can resolve individual particle interactions, a key requirement for time-of-flight (ToF) analysis and reaction cross-section studies. Consequently, detection methods that integrate over all interactions within their active volume, such as passive neutron detectors (e.g., bubble detectors[36]), retrospective activation methods[37], or the typical current-mode operation of scintillator detectors[38] that are commonly and successfully employed in LDNS experiments for neutron counting and bulk spectrometry, are incompatible with this specific requirement. Thus, a single-event fast neutron detection system, capable of operating in the harsh environment of an LPA, must be established and, what is more difficult, very shortly after the laser shot.

(2) To achieve sufficient statistics for reliable neutron time-of-flight analysis in single-event experiments, individual detector signals must be accumulated over multiple laser shots. The precise number of laser shots required is highly variable and experiment-specific, depending on factors such as the detector's intrinsic efficiency, the solid angle coverage, the target nuclear cross-section, and the desired energy resolution. For instance, ~200 laser shots are sufficient for the demonstration experiment presented herein, whereas more than $10^5$ proton beam pulses at the CERN n_TOF spallation source have been necessary recently to accurately measure the $^{12}C(n,p/d)$ reactions up to 25 MeV[39]. This inherent requirement for accumulation dictates the need for stable, high-repetition-rate source operation and shot-to-shot monitoring of the experimental conditions to ensure data integrity.

(3) Accurate background characterization and subtraction, which is especially critical in LDNS environments. Unlike conventional accelerator-based sources, laser-plasma interactions generate an intense, prompt, and mixed background—including photons, electrons, ions, and scattered or secondary neutrons—whose properties are not yet fully understood or standardized.

These capabilities are demonstrated simultaneously for the first time in this work (see Refs. 40,41 for more details). The experiment discussed herein established the feasibility of studying fast neutron-induced nuclear reactions via neutron time-of-flight using a petawatt-class, ultrashort-pulse laser source operating in a stable multi-shot mode.

A suite of dedicated particle diagnostics enabled shot-to-shot monitoring of neutron, electron, proton, ion, and photon production, demonstrating the required source stability. In parallel, Monte Carlo (MC) simulations were used to model the influence of background radiation components at the detector position, which were also measured. A high-performance diamond detector was placed at a flight path of only 1.5 m, maximizing neutron flux while maintaining sufficient neutron energy resolution.

Signals from individual fast neutron-induced reactions were successfully detected over hundreds of laser shots. The measured time-of-flight spectrum is consistent with MC predictions based solely on the LPA beam diagnostics, overcoming long-standing limitations related to detector suitability, source stability, and background complexity in this challenging environment.

## Results

### Design and implementation of a time-of-flight setup for an LDNS

Traditional accelerator-based neutron time-of-flight facilities maintain a distinct spatial separation between the neutron source and the detection apparatus. Thick walls equipped with collimators and neutron beam guides define the neutron flight path towards the measuring stations and suppress photon and neutron scattering backgrounds from the neutron production site. This is essential for ensuring accurate and reliable experimental results, but, for now, it remains impractical with current LDNSs, as their multi-purpose laser interaction rooms are not yet optimized for neutron time-of-flight measurements. This implies a short standoff distance of the neutron detection systems not only to the neutron production target, commonly referred to as the *catcher*, but also to the laser-plasma interaction point, or *pitcher*. The latter emits a strong pulse of electromagnetic radiation (EMP) in the entire spectrum (THz, optical, X-rays, and hard $\gamma$-rays) as well as particles (electrons, protons, heavy ions) across a large solid angle, some of which are energetic enough to produce neutrons and photons not only in the catcher but in other components of the experimental setup and the laser interaction chamber. To address these challenges, a streamlined setup was employed, focusing on three key tasks, as illustrated in Figs. 1–3: (1) Characterizing the laser-plasma interaction to feed MC simulations for (2) predicting neutron generation channels and detection rates in an optimized detector and shielding configuration; and (3) continuous shot-to-shot monitoring of particle beams generated in the LPA process to enable data accumulation, as required by high-precision nuclear physics experiments.

The experiment utilized the DRACO Petawatt (PW) laser at Helmholtz-Zentrum Dresden-Rossendorf. The laser system is capable of delivering pulses of up to 18 J (on target) with a pulse duration of 30 fs[42]. These pulses were directed on the pitcher target at an oblique incidence angle of 50° using an f/2.3 parabolic mirror, achieving a peak intensity of ~$5 \times 10^{21}$ W/cm$^2$. To improve the laser pulse's temporal contrast, and thereby optimize acceleration performance and stability, a single plasma mirror setup was employed[20]. This, however, restricts the experiment's repetition rate to about one shot every ten seconds and the daily shot count to approximately 200. The experiment employed plastic foils with a thickness of about 250 nm as pitcher targets, with protons and carbon ions being the most abundantly accelerated species. These ions were emitted along the normal to both the rear and front sides of the target through the well-established Target Normal Sheath Acceleration (TNSA) mechanism[43].

The ion beam generated at the target's rear was characterized using a suite of detectors, each based on different detection principles. These measurements were used to determine the contributions responsible for neutron generation in the catcher (details in the "Methods" section). A Thomson Parabola Spectrometer (TPS) was placed along the target's normal to analyze particle spectra from the rear surface of the pitcher (forward direction, fwd) with high energy resolution. It featured a microchannel plate readout for cut-off energy detection and a $Gd_2O_2S$:Tb (GOS, on plastic support layer) scintillating screen for the absolute quantification of particle numbers[44]. To complement the spectral analysis, another GOS scintillating screen was positioned ~10 cm behind the pitcher target to measure lateral beam profiles of accelerated protons above a discrete threshold energy of 40 MeV. Alongside the TPS, the profiler was employed to optimize the beam daily, following the protocol in Ref. 20. A magnetic spectrometer adapted from[45] was used to estimate the proton cut-off energy coming from the front side (laser incident side) of the pitcher (backward direction, bwd) by differentiating between protons and heavier ions with aluminum filters. For selected shots, stacks of radiochromic films (RCFs) were placed to map the energy-resolved spatial dose distribution delivered by the proton beam. Electrons and bremsstrahlung radiation were measured along the laser propagation direction. This was achieved by combining a magnetic electron spectrometer

equipped with a GOS screen readout inside and an imaging-plate (IP)-based bremsstrahlung calorimeter[46,47] placed outside the vacuum chamber. The calorimeter contained a sequence of absorber plates of different materials with increasing thickness. These plates enabled the formation of a secondary particle shower, which was recorded by the IPs as active layers.

Figure 1b summarizes the particle beam spectra retrieved from the detector suite and used as input for the MC simulations. The proton energy spectrum directed towards the catcher is shown in the top panel. It exhibits the typical quasi-exponentially decaying spectral shape with well-defined cut-off energy of about 50 MeV, surpassing by far the 1.9 MeV, 2.2 MeV and 4.2 MeV thresholds of the $^7$Li, $^{63}$Cu, and $^{65}$Cu (p,n) reactions. With yields of about $5 \times 10^{11}$ protons/shot for energies above 2 MeV, the setup enables prolific fast-neutron generation. Regarding carbon ions (not shown), cutoff energies of $E_{max} \sim$ $13\,MeV/u$ were measured, and a total number of ~$3 \times 10^9$ ions (>1 MeV) was estimated for use in the simulations by scaling the energy deposition measured in the detector according to the stopping power difference between protons and carbon ions; a conservative estimate was adopted, as the detector was not absolutely calibrated for these species. The contribution of oxygen ions was not examined but deemed negligible for this analysis. The backward-emitted spectrum from the front face of the pitcher is reconstructed using the RCF data from the rear side and the cutoff energy of about 35 MeV measured with the magnetic spectrometer (not shown), assuming a TNSA scenario[48]. During the laser-plasma interaction, electron and bremsstrahlung emission are strongest in the laser-forward direction, where respective energy spectra were measured (middle and bottom panels of Fig. 1b). To estimate their contribution to the neutron background, the simulations assumed isotropic emission characteristics and conservatively scaled the absolute numbers of electrons and photons, based on experimental measurements.

Neutrons were generated using two different catcher configurations: 3 mm thick Cu and 10 mm thick LiF. Inside the catcher material, fast neutrons are mainly produced via (p,n) reactions and emitted isotropically with MeV energies, exhibiting an exponential behavior inherited from the incident proton beam. To study neutron-induced nuclear reactions—with time-of-flight measurements being the most demanding case—it is essential to detect individual interaction events and extract relevant observables, such as timing and deposited energy. This requires detectors capable of operating in single-event mode with high temporal resolution. A detector suited to this task must combine fast response (to resolve signals separated by nanoseconds), radiation hardness (to withstand the prompt electromagnetic and particle emissions), and insensitivity to gamma and X-ray backgrounds (to avoid blinding or pile-up effects). Additionally, low intrinsic detection efficiency is beneficial to minimize pile-up in high-flux conditions.

To meet these requirements, a single-crystal diamond detector was selected and deployed in time-of-flight configuration (marked in light blue in Fig. 1a). These detectors are known for their sub-ns response time, robustness under radiation, low sensitivity to photons, and proven suitability for neutron-induced reaction studies in harsh environments. Indeed, these detectors have previously been employed successfully in ToF nuclear physics experiments involving neutron-induced reactions, such as (n,$\alpha$) and fission[49,50]. Their demonstrated performance in demanding radiation environments—such as the n_TOF-NEAR station at CERN[51] and the planned neutron diagnostics at ITER[52,53]—supports their suitability for use in LPA sources as well.

In previous LDNS experiments, detection systems have mostly operated in current or integral mode (relying on a fast neutron scintillator signal proportional to the total instantaneous rate of interactions) or relied on passive detectors such as activation foils or bubble dosimeters. While these bulk-integrated approaches are valid for characterizing the source, providing essential data such as total

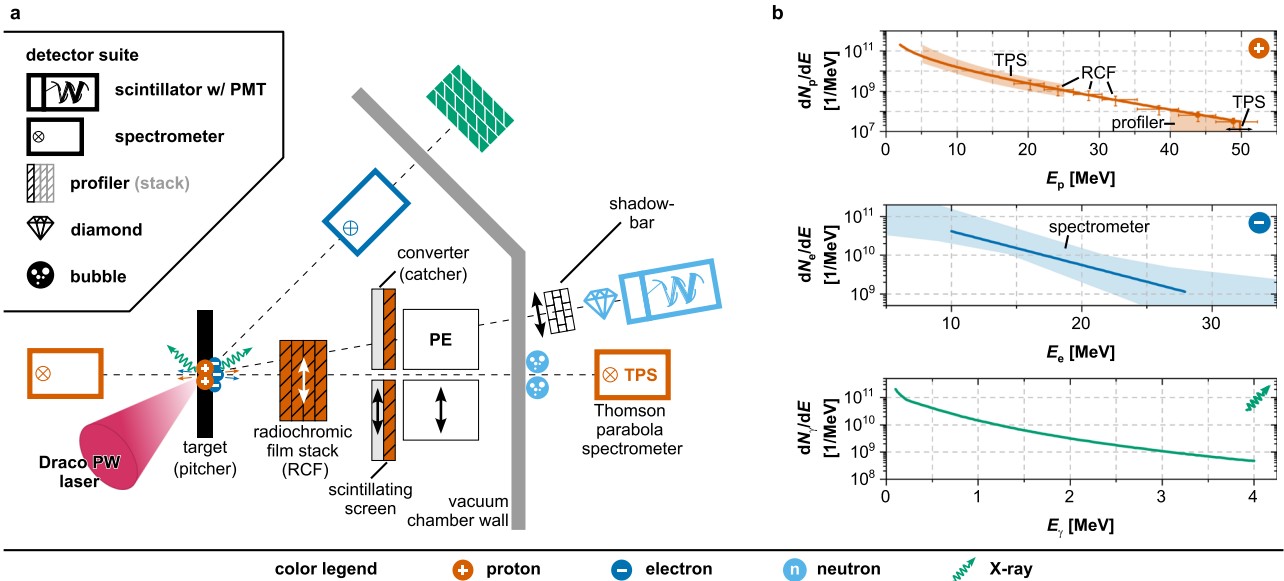

**Fig. 1 | Experimental setup and simulation input. a** Experimental setup. The DRACO PW laser is obliquely incident on the target. Proton/ion diagnostics (orange) are aligned along the target normal axis and comprise beam profilers (hatched boxes) and magnetic spectrometers. The electron spectrometer (dark blue) and bremsstrahlung calorimeter (green) are aligned along the laser forward axis. Diamond and neutron time-of-flight scintillators coupled to a photomultiplier tube (PMT) (light blue symbols) are aligned at ~30° from the laser axis. Bubble detectors (light blue) and radiochromic film (RCF) stacks were placed for dedicated shots. Behind the target, the neutron converter, the PE beam dump, and the shadow-bar in front of the diamond detector could be placed or removed as desired. **b** Simulation input. Solid lines in the panels represent interpolations of the absolute measured spectral data for protons (top), electrons (middle), and bremsstrahlung (bottom) used as input for the MC simulations. The shaded areas refer to the data regions monitored on a single-shot basis with the corresponding detectors, as indicated by the symbols (see also Fig. 3).

neutron yields and source stability estimates, they fundamentally lack the temporal resolution and event-specific fidelity necessary for time-of-flight measurements and the analysis of individual neutron reaction kinematics as pursued in this work. Furthermore, in LDNS measurements, the scintillators are normally surrounded by massive shielding, as in the experiment presented herein, which heavily affects the detector response to both neutrons and photons.

Nonetheless, such conventional detectors remain useful as auxiliary diagnostics. For shot-to-shot neutron monitoring, a standard EJ-232Q liquid scintillator[38] coupled to a PMT was placed at 3.6 m from the source and shielded by up to 15 cm of lead and polyethylene. Although limited to providing an integrated signal proportional to the total neutron production, this is sufficient to assess the relative stability of the source on a shot-to-shot basis. Additionally, passive bubble detectors from Bubble Technology Industries[54] were used to determine the absolute fast-neutron yield.

**Secondary neutron sources.** Contrary to conventional accelerator-based sources, particles other than protons are also produced in the laser-pitcher interaction and are emitted in different directions. The impact of these is strongly dependent on the specifics of the experimental setup and, to the best of the author's knowledge, has not been addressed in current LDNS literature, where the focus often remains on the primary proton beam. In this work, their contribution was assessed using MC simulations. As illustrated in Fig. 2a, it turns out that a small but non-negligible fraction of the neutrons originates from reactions other than proton-induced interactions in the catcher. This secondary contribution is dominated by carbon ions for neutrons above a few MeV, and electrons for neutrons below 1 MeV.

**Photon background.** Since all charged particles are absorbed either by the catcher, the wall, or the shielding, only direct or secondary photons can reach the detector and induce additional signals to those from neutrons. Their contribution at the diamond detector position is

displayed in Fig. 2b alongside the simulated neutron yield: the photon background, produced predominantly by electrons, is only significant at short flight times, i.e, in the region just before the arrival of the highest-energy neutrons (over 30 MeV).

**Scattered (indirect) neutrons.** A crucial aspect in ToF experiments is the fraction of neutrons reaching the detector indirectly after being scattered or produced in secondary processes by the surrounding elements of the setup. Kinetic energies and arrival times of the neutrons reaching the diamond detector were simultaneously scored, and the corresponding heatmap plot is displayed in Fig. 2c, on top of which the dashed red line indicates the expected relationship for neutrons traveling from the catcher to the detector without interacting along the way. All signals far from this kinematic region can be attributed to scattered or secondary neutrons, i.e., neutron background. The direct and background components can be distinguished better in the inset: all signals differing from the expected behavior by more than the 3 ns duration of the ion pulse can be attributed to background. According to the simulations, in our setup, the fraction of direct neutrons at the diamond detector is still dominant over the background at 65%. In comparison, the fraction of signal versus background amounts to 48% for the bubble detector (closer to the catcher but surrounded by structural material, and more sensitive to background due to its relatively flat efficiency down to low neutron energies), and only 9% for the scintillator (further from the catcher, surrounded by massive shielding and close to the hall walls).

These observations highlight the complexity and limitations of using scintillator setups and integrating detectors as bubble detectors for neutron yield characterization or comparative studies between different LDNS, when discriminating signal origins is essential.

The simulation results in Fig. 2 illustrate the key design trade-offs to consider when optimizing a time-of-flight setup. A shorter flight path generally increases the neutron flux while reducing the neutron energy resolution. In this context, the neutron flux in the detector

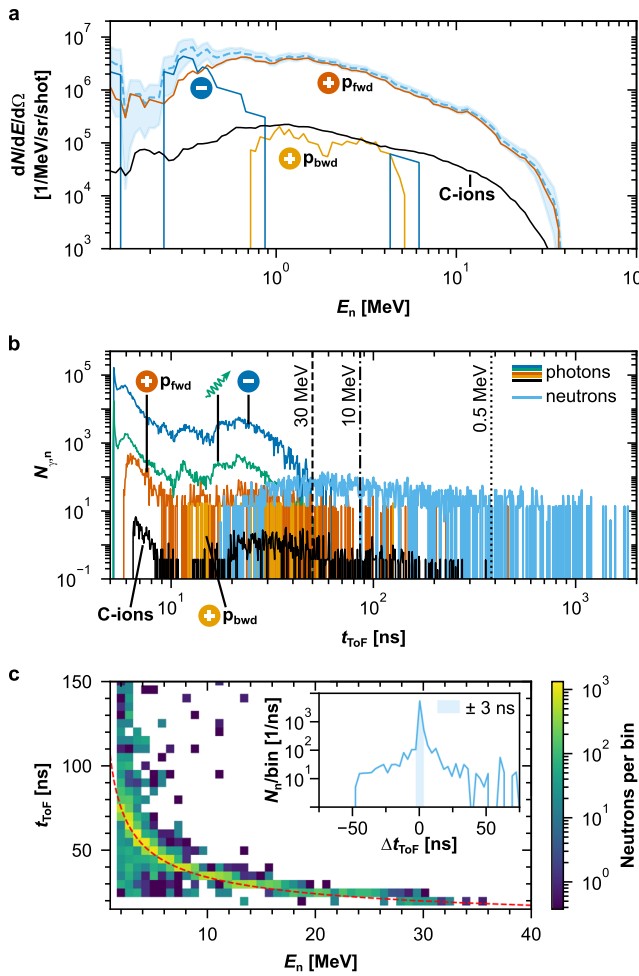

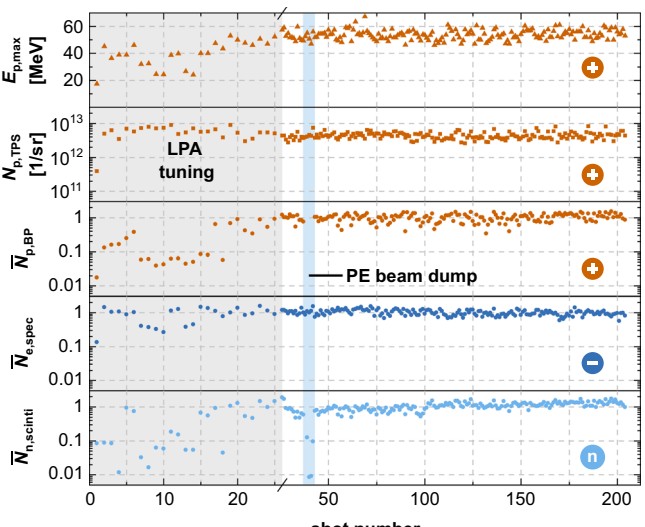

**Fig. 3 | Shot-to-shot beam monitoring.** Shot-to-shot stability of protons (TPS cut-off energy $E_{p,\,max}$, integrated proton yield from TPS $N_{p,TPS}$ (5 MeV to 25 MeV) and beam profiler $\overline{N}_{p,BP}$ (>40 MeV)), electrons $\overline{N}_{e,spec}$ as marked in Fig. 1b, and the resulting neutron production (integrated scintillator yield $\overline{N}_{n,scinti}$) for 204 consecutive laser shots using the Cu-plate catcher configuration. Shaded areas mark the shot range where the laser-plasma accelerator was tuned (shots 1–26) and the PE-block was used as a beam dump (shots 38–42). The strong dip in $N_{p,BP}$ during LPA tuning is because proton energies dropped below the energy threshold of the beam profiler. The unshaded region represents shots with optimized neutron performance, stable within 26% standard deviation. The quantities of the lower three rows are normalized to the mean values within the optimized range.

**Fig. 2 | MC simulation results for particles arriving at the diamond detector.**
**a** The flux of neutrons as a function of neutron energy according to their origin. Forward-emitted protons ($p^+_{fwd}$) dominate neutron production in the high-energy region. The non-continuous spectrum for neutrons generated by electrons is an artifact of insufficient statistics. The light blue dashed line corresponds to the sum of the individual neutron production channels. The shaded area represents the statistical uncertainty of the simulation results for a given energy bin. **b** Time of arrival of prompt photon radiation fields (predominantly from bremsstrahlung from the source and bremsstrahlung produced by electrons elsewhere) vs. time of arrival of neutrons. A shorter flight path would imply overlap between the arrival of photons and high-energy neutrons. The vertical dashed/dotted lines mark different neutron energies. **c** Correlation between the time-of-flight and energy of the neutrons. All events far from the red-dashed line, representing primary neutrons, correspond to neutrons scattered or produced elsewhere by secondary sources other than in the catcher target. The inset displays a histogram of the difference between the neutrons' arrival time at the detector and the expected arrival time for the direct path. The expected times were calculated from the neutrons' energies. Only the neutrons with differences of less than 3 ns (width of the initial ion pulse, shaded area. See inset in Fig. 5a) are considered primary neutrons produced in the catcher.

needs to be high enough to provide a sufficient count rate to allow for a measurement of the reaction yield, given the mentioned limit in the number of laser shots per day, but avoiding both signal pile-up and superposition of the γ-flash and the neutron signals. In this work, at the chosen distance, the simulations predict a reasonable energy resolution—on the order of a few percent, assuming a neutron production time of approximately 3 ns—with only minimal overlap between photon and neutron events. The expected count rate in the diamond detector is a few signals per laser shot, which is sufficient to keep pile-

up under control while still providing adequate statistics within the ~ 200 shots-per-day limit imposed by the use of the plasma mirror.

## Shot-to-shot monitoring

Stability and reproducibility of the laser acceleration performance are essential for multi-shot nuclear physics experiments at LDNS, given that such sources exhibit fluctuations in energy and intensity. This has been a major concern for multi-shot applications of LPA, such as radiation therapy for cancer research, and can only be addressed through reliable shot-to-shot monitoring.

As illustrated in Fig. 3, all employed diagnostics—including proton cut-off energy, proton yield, and high-energy electron production—indicate stable performance of the LPA source and subsequent neutron production under optimized conditions throughout a single shot day.

Quantitatively, the relative standard deviations (RSD) obtained across the 193 consecutive laser shots with the Cu target (unshaded region in Fig. 3) were: 7% for the proton cut-off energy ($E_{p,\,max}$ = 54(4) MeV), 24% for the proton yield ($N_{p,TPS}$ = 4(1) × 10^{12} p/shot), 26% for the proton beam profiler signal, 19% for the high-energy electron signal, and 26% for the neutron signal measured by the scintillator detector. These values demonstrate that, despite the inherent shot-to-shot fluctuations in laser-plasma acceleration, the system maintained adequate stability to enable the accumulation of statistically significant single-event neutron data.

Two dedicated subsets of shots indicated in Fig. 3 further demonstrate that the scintillator-based neutron diagnostic can effectively assess LDNS stability: (i) inserting a PE beam dump that fully absorbs the ion beam (eliminating neutron production), which leaves only a residual background signal on the scintillator, and (ii) during the tuning phase of the LPA (for best focus and optimized spectral phase, following the protocol in[20]), where the neutron signal closely follows the evolution of the primary particle beams.

It is important to note that there is no widely established benchmark for these RSD values in high-power laser plasma acceleration experiments, as traditionally such experiments have been conducted in a single-shot regime with extensive preparation for each individual shot[23]. Typically, only one or very few shots per day were performed. Therefore, the significance of our reported RSDs lies not necessarily in their absolute magnitude, but rather in the ability to maintain consistent and repeatable fluctuations across hundreds of consecutive shots. This stable, multi-shot operation enables effective shot-by-shot monitoring, allowing the identification and exclusion of non-optimal shots or experimental drifts, which is essential for accumulating statistically meaningful data in nuclear physics applications.

### Single-event-based fast neutron time-of-flight spectrometry

In this work, a fast neutron and particle detector with low efficiency and low gamma sensitivity−a single-crystal diamond detector from *CIVIDEC Instrumentation*[55]−was successfully operated at only 1.5 m from the LDNS target. This short flight path, chosen to maximize the

neutron flux, represents a challenging scenario due to the intense prompt electromagnetic and particle background generated by the laser-plasma interaction. Despite these conditions, the detector was able to register signals from individual interactions of fast neutrons with the carbon atoms in the detector material[50], demonstrating the feasibility of single-event spectrometry in the immediate vicinity of a petawatt-class neutron source.

A trace from the diamond detector corresponding to a single laser shot is shown in Fig. 4a, obtained after optimizing the acquisition conditions. The plot illustrates the trace before (raw trace, gray) and after applying quality improvements by subtracting the signal of the γ-flash, the corresponding baseline undershoot, and the EMP-induced noise (green trace, details in "Methods"). Still, a minimum time-of-flight threshold of 20 ns remains necessary in the analysis, rejecting all signals with shorter arrival times because they overlap with the residual dominant γ-flash structure (indicated by the shaded gray area in Fig. 4a). This corresponds to a maximum detectable neutron energy of 30 MeV, consistent with the expectations from the Monte Carlo

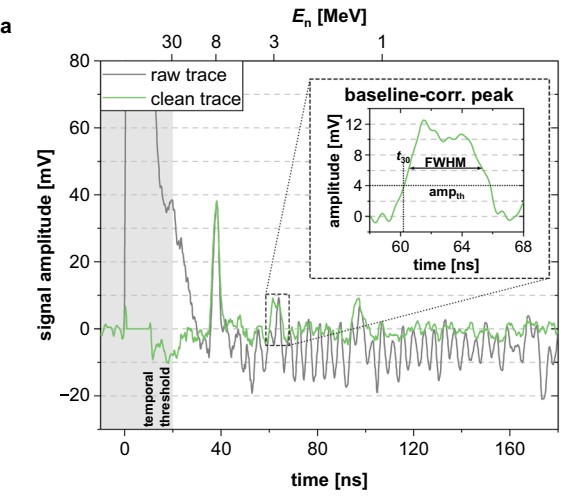

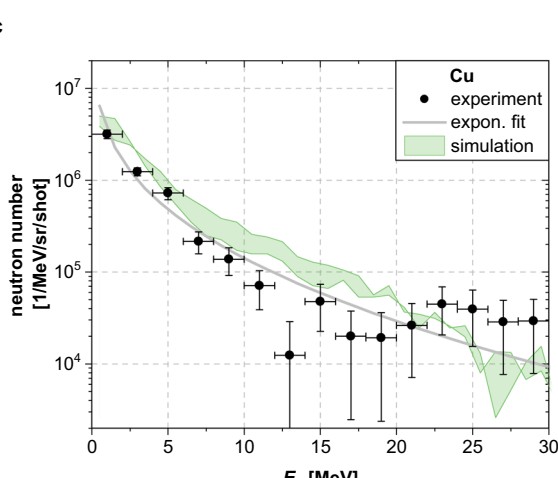

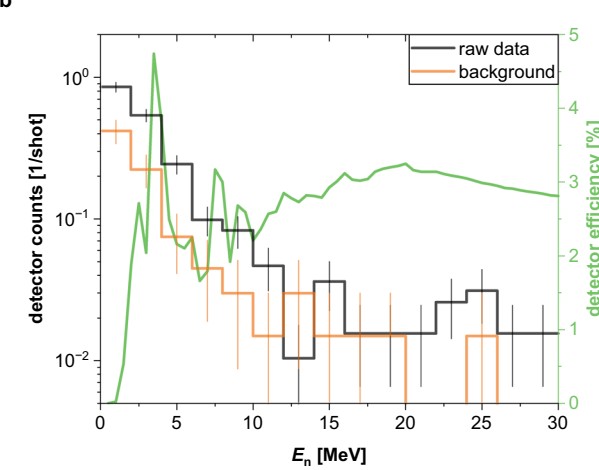

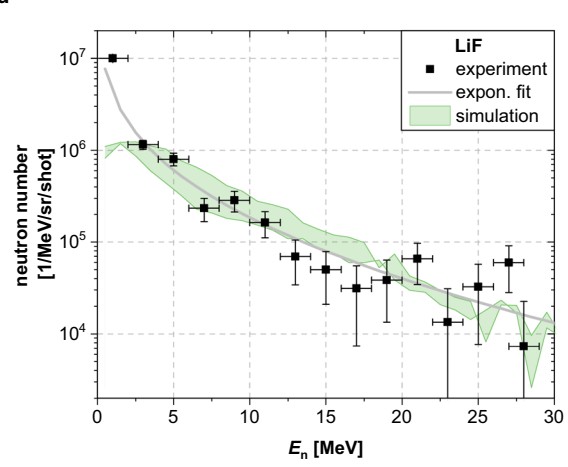

**Fig. 4 | Neutron time-of-flight measurements with the diamond detector.**
**a** Exemplary trace from the diamond detector before and after processing. The inset shows the notation of the signal characteristics for a selected single neutron signal. **b** Neutron energy histograms accumulated over 67 laser shots taken with the shadow-bar in place (background) and 193 shots taken without it (raw) for the p-Cu pitcher-catcher configuration, showing the effect of the background. In green, the diamond detection efficiency is plotted considering the established 400 keV deposited energy threshold. **c**, **d** Neutron yield obtained for Cu(p,n) (**c**) and Li(p,n)

(**d**) obtained via Bayesian ON/OFF analysis for background compensation and taking into account the detector efficiency from (**b**). Experimental uncertainties represent the FWHM of the resulting posterior distributions propagated to the yield units. The exponential fit of the experimental data is shown as a solid gray line and corresponds with Eq. 11 in Ref. 63. Simulations are displayed as a band between Geant4 (lower limit) and PHITS (upper limit), estimating the uncertainty associated with the simulations.

simulations shown in Fig. 2b. The remaining EMP-related fluctuations were still comparable to the amplitude of the neutron signals, necessitating the development of a dedicated pulse Shape Analysis (PSA) algorithm. This algorithm identifies and analyzes neutron-induced reaction signals in the diamond, considering as candidates positive signals with an amplitude larger than two times the noise standard deviation, equivalent to 4 mV amplitude. These were analyzed and characterized by (see inset) amplitude, integral, width (FWHM), and timestamp (time corresponding to 30% of the amplitude in the rising slope), i.e., the time-of-flight, which is then converted into the corresponding kinetic neutron energy. Care was taken upon the occurrence of multiple-peak structures, which were statistically examined to discard noise peaks superimposed on actual signals that would then exceed the threshold.

To determine the neutron spectrum generated at DRACO PW, signals from individual neutron interactions were measured using the diamond detector and accumulated over hundreds of laser shots for two catcher materials: LiF (160 laser shots, with ~ 2.3 signals/shot after background subtraction) and Cu (193 laser shots, with ~ 1.2 signals/shot after background subtraction). Lithium was the natural choice, as it is commonly used in accelerator-based facilities operating in similar energy ranges[56–58]. Copper, on the other hand, was selected as an alternative configuration to support the validation of the simulation results.

The corresponding time-of-flight spectrum from each catcher was converted into neutron energy distributions after background subtraction and correction for the neutron detection efficiency. For background estimation in the detector due to the indirect neutrons discussed before, a shadow-bar (10 cm wide and 40 cm long block of borated polyethylene) was inserted in the line of sight between the neutron source and the detector. This shadow-bar, although limited in size because of the lack of space, acted as an efficient neutron shield[59]: A Geant4 Monte Carlo simulation showed that it absorbed or scattered 99,9% of the incident neutrons, and as much as 99,0% if only the more penetrating high energies (> 10 MeV) are considered. Ideally, an additional measurement without a catcher would help in understanding the nature of the background not related to neutrons. This was attempted using a thick polyethylene catcher as a dummy, but with proton beam energies exceeding 20 MeV, even the $^{12}C(p,n)$ reaction channel (Q = −18 MeV) was open, and the measurement produced a measurable neutron field (see shot ~ 30 in Fig. 3). In Fig. 4b, the neutron energy histograms accumulated over 67 laser shots with (background) and 193 laser shots without (raw) the shadow-bar are compared for the Cu-catcher configuration. According to the shadow-bar method, neutron scattering accounts for 40% to 50% of the signal. This is consistent with expectations, considering the limited laboratory space and shielding constraints. The measured contribution matches the prediction of the simulation presented in Fig. 2c within a 10% (see the section on "Scattered (indirect) neutrons"). The neutron detection efficiency (green line in Fig. 4b) depends on the neutron energy and the signal amplitude threshold, in which the structures visible around 5 MeV correspond to the resonances in the cross-sections for all possible $^{12,13}C(n,*)$ reactions[50]. The efficiency curve is obtained based on Geant4 simulations[60] of the detector, considering mono-energetic neutrons in the range of 0 MeV to 50 MeV and counting the events with deposited energy above the detection threshold of 400 keV. This value was selected as a compromise: sufficiently low to maximize the limited statistics, yet high enough to remain above the noise level.

Given the limited statistics, a Bayesian framework was employed for the data analysis. The direct neutron energy distribution was estimated using an ON/OFF measurement approach[61,62], where the ON data represent the combined source and background signals, and the OFF data correspond to background measurements taken with the shadow-bar in place. For each energy bin, a Poisson likelihood was used to construct the posterior probability distribution of the neutron signals.

After background subtraction and detection efficiency correction, the neutron energy distributions are then converted into neutron yield in the conventional units of n/sr/MeV/shot by considering the number of laser shots delivered, the solid angle covered by the diamond, and the bin size of the histograms. The neutron yields for the Cu and LiF catchers are shown in Fig. 4c, d. In both cases, the neutron spectrum exhibits nearly exponential behavior, consistent with the TNSA shape of the laser-driven proton spectrum[63].

For neutrons with energies above 100 keV, yields of ~ $2 \times 10^8$ n/shot were measured for both Cu and LiF catchers. Above 1 MeV, the yields decrease to $9 \times 10^7$ n/shot and $1 \times 10^8$ n/shot, respectively. This nearly 30% difference between catchers is consistent with simulation predictions and can be attributed to the higher threshold of the Cu(p,n) reaction. The maximum neutron energy measured in both cases is ~ 30 MeV, limited by the shortest time-of-flight accessible after the γ-flash and by reduced statistics at higher energies. Nonetheless, neutron production up to 50 MeV is expected for the proton energy range of the primary beam (cf. Fig. 1b).

These neutron yields per shot measured by the diamond detector are consistent with results obtained using conventional methods, such as bubble detector measurements, matching their order of magnitude and falling within a factor of 4 (see Ref. 64).

Furthermore, these experimental results for both catcher materials (LiF and Cu) were subjected to a validation against Monte Carlo (MC) simulation predictions from Geant4 and PHITS using nuclear data from the TENDL-2019[65] and JENDL-4[66] evaluated libraries, respectively. The uncertainty band between the codes arises from differences in the particle transport, nuclear interactions methods and cross-section data employed by Geant4 and PHITS. These discrepancies between different codes in modeling high-energy particle-induced reactions and subsequent neutron transport are well-known in the nuclear physics community, and observed in other neutron production calculations, for instance at the n_TOF spallation facility[67]. Therefore, the band shown in Fig. 4c, d can be considered only an estimate (with its range defining a lower limit for the associated uncertainty) of the neutron spectrum, against which the experiment is compared.

This cross-validation, which included matching the total neutron flux, the spectral shape, and the relative contribution of the neutron scattering background (measured via the shadow-bar technique), demonstrated a strong level of agreement, falling within 20% to 50%. This degree of fidelity is noteworthy considering that the limited accuracy of the simulations, that each data set was obtained from fewer than 200 laser shots —limiting the available statistics— and that daily fluctuations in proton yield were already of comparable magnitude.

Overall, although no explicit nuclear reaction was investigated in this experiment, the registration and processing of single neutron-induced signals generated within the diamond detector already correspond to $^{12,13}C(n,*)$ reactions produced within the detector material itself. In addition, the ability to identify individual signals and extract well-defined observables—such as time, amplitude, and FWHM (related to the incident energy, deposited energy, and signal shape, respectively)—opens the door to pulse-shape discrimination analysis and enables the exploration of correlations between these observables for deeper insight into the reaction processes of interest. These findings demonstrate the feasibility of studying neutron-induced nuclear reactions via time-of-flight using a PW laser-driven neutron source.

## Discussion

The results presented herein allow discussing the DRACO LDNS in the landscape of ion accelerator-based neutron sources for nuclear reactions. Considering the main features of a neutron time-of-flight facility are neutron energy range, energy resolution, and neutron flux, either per pulse or average, LDNSs offer the potential of unprecedented

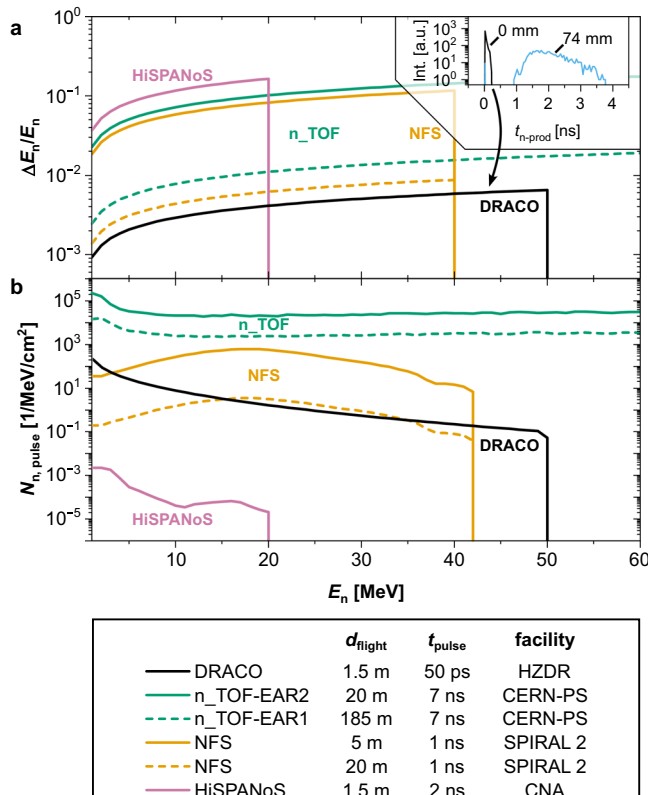

**Fig. 5 | Comparison with established accelerator-based neutron sources.** Performance of the DRACO LDNS (calculated from Geant4 simulations neglecting the unnecessary 74 mm separation between pitcher and catcher) and conventional ion accelerator neutron beam facilities. **a** The exceptionally short ion pulse generated by the laser (see inset) results in a better resolution at only 1.5 meters than other state-of-the-art facilities featuring flight paths as large as tens and even hundreds of meters. **b** The neutron yield per pulse varies enormously between facilities, featuring the DRACO LDNS orders of magnitude higher values than current compact accelerator-based neutron sources (CANS, i.e., HiSPANoS), but, as NFS, orders of magnitude lower than the spallation-driven ToF facilities.

neutron flux per shot and energy resolution thanks to the high-power and ultra-short duration of the laser pulse.

Regarding the excellent neutron energy resolution, it is related to the fs duration of the laser pulse that produces ion pulses lasting only a few ps[68]. According to Geant4 Monte Carlo simulations, the most energetic of these ions, mainly responsible for the neutron production, spend only a few tens of ps traversing the catcher target, producing a neutron burst of the same temporal scale, up to ~50 ps in the current experimental setup. In this work, due to the 74 mm separation between pitcher and catcher, required for parallel characterization of the primary sources, the neutron pulse duration was further broadened to a few ns; however, this distance can be easily reduced to near contact to significantly shorten the neutron pulse duration, and thus it is neglected in the discussion that follows. As illustrated in Fig. 5a, a 50 ps neutron pulse at DRACO PW (best-case scenario) yields an energy resolution better than 1% over the entire energy range using the flight path of 1.5 m. This performance surpasses that of current state-of-the-art ion accelerator-driven ToF facilities, including compact (HiSPANoS@CNA, 2 ns bursts)[58], medium-scale (NFS@SPIRAL2, 1 ns bursts)[57], and large-scale setups (n_TOF@CERN, 16 ns)[14,15], many of which require flight paths of tens to hundreds of meters. This means that the combination of fast state-of-the-art detectors (10-30-100 ps resolution for plastics, LGADs, and diamonds[69-71]) with the ToF setup associated with an LDNS for nuclear physics can remain extremely

compact, in line with the promised advantages of laser-driven accelerators.

Regarding neutron source intensity, Fig. 5b shows the neutron yield per pulse for the same facilities as discussed above. Notably, there is an eight-order-of-magnitude difference between a compact source using a modest 3 MV tandem accelerator (HiSPANoS@CNA) and the record-high flux achieved via 20 GeV proton spallation at CERN n_TOF. In this landscape, the intensity per pulse of the DRACO LDNS is comparable to that of the NFS facility at SPIRAL2 driven by a 40 MeV deuteron beam, with neutron flight paths of 5 and 20 m.

Achieving statistically significant results is essential for the success of nuclear physics experiments at LDNSs. To fully harness the potential of LDNSs, PW-class lasers operating at repetition rates of several Hz are essential, considering that low repetition rates are only common in spallation sources, while lower energy accelerator-driven neutron sources (for instance HiSPANoS or NFS) operate in the MHz regime. In this regard, results such as the recent demonstration of 10 Hz operation at ELI-ALPS for comparable beam powers[72] are highly promising. Access to high repetition rates also demands advanced pitcher target solutions that enable a significantly larger, or even unlimited, number of laser shots. Encouraging progress is being made with concepts such as automated target holders[73], tape targets[74], self-replenishing cryogenic jets[75], and liquid leaf jets[76,77]. Additionally, laser ion acceleration efficiency is advancing rapidly: for example, recent experiments at the DRACO facility achieved higher proton fluxes and proton energies exceeding 100 MeV, notably without using the shot-rate-limiting plasma mirror[78].

These developments, together with our proof-of-concept study on single-event-based neutron spectrometry, support the conclusion that LDNSs—fueled by rapid progress in laser-plasma acceleration—are emerging as viable and complementary tools for neutron-induced nuclear reaction studies, particularly in scenarios where short pulse durations and high instantaneous fluxes provide critical advantages.

Having demonstrated the feasibility of measuring neutron-induced reactions, the accuracy at reach is still to be studied. Thus, the next necessary step in this endeavor consists of measuring a nuclear reaction with a well-known cross-section. Assuming a 10-fold neutron production[78] and 10 Hz (as at ELI), cross-sections in the order of 1 barn could be measured within a 3% statistical uncertainty (1500 counts would be accumulated for each sample in 1 MeV bins around 10 MeV) in only one week of beam time employing a four-fold 10 × 10 mm² mosaic diamond detector loaded with different samples. This should be enough to assess the accuracy at reach for nuclear reaction studies with the LDNS approach discussed herein. An interesting choice of samples and reactions for such benchmark experiment could be fission on ²³⁵U and ²³⁹Pu in the spirit of the very recent experiment of the NIFFTE Collaboration[79] at LANL for measuring the ²³⁹Pu/²³⁵U fission cross-section ratio with an unprecedented accuracy of just 1%, Such measurement represents the state-of-the art for neutron reaction measurements in terms of accuracy, wealth of observables and impact of the results, being therefore an excellent test bench for forthcoming measurements at a LDNS.

## Methods
### Laser-driven neutron source
Experiments were performed at the DRACO PW laser at the Helmholtz-Zentrum Dresden-Rossendorf. DRACO PW is a double-CPA Ti:Sa laser system, providing 30 fs (FWHM) laser pulses with a central wavelength of 810 nm (50 nm bandwidth FWHM). The pulses carried an energy of ~18 J and were focused onto thin (~250 nm) Formvar plastic foils ($C_5H_8O_2$, $n_e = 230 n_c$ when fully ionized) by an f/2.3 off-axis parabolic mirror to a FWHM spot size of 2.6 μm, yielding peak intensities of $5.4 \times 10^{21}$ W/cm². Plasma mirror (PM) cleaning was applied to improve the temporal contrast by almost four orders of magnitude, essentially removing all remaining prepulses and limiting the ionization and

plasma dynamics to the last ps before the peak of the laser[20,80–82]. PM operation limited the shot rate to about one shot every 20 s and the total number of shots to about 200 shots per day. The work presented herein corresponds to neutrons generated using two different catcher options (10 mm thick LiF and 3 mm thick Cu) on two separate days. The complete experimental campaign took place over several days, which were devoted to the optimization of the LPA and detection systems

### Particle diagnostics

The spatial proton intensity distribution was measured using calibrated radiochromic film (RCF) stacks in several reference shots. A Thomson Parabola spectrometer (TPS) placed at 45° was used to measure ion spectra for every laser shot. A small fraction of the ion beam emitted from the target was delivered by passing through the 4 mm diameter hole in the catcher and a pair of small pinholes towards the TPS. This resulted in a set of two parabolic traces, one recorded with a microchannel plate in imaging configuration to assess the cut-off energy with high spatial resolution and high dynamic range, and the other used a calibrated $Gd_2O_2S$:Tb (GOS) scintillating screen[44] for absolute proton number determination in the proton energy range of 5 MeV to 25 MeV. The TPS and RCF data, with remarkable agreement, indicate that a total of $5 \times 10^{12}$ protons/sr/shot to $6 \times 10^{12}$ protons/sr/shot are produced by each laser pulse. Considering the beam divergence of around ~20°, this corresponds to a proton beam intensity of $5.6 \times 10^{11}$ particles/shot (spectrally integrated for > 2 MeV). The predominant heavy ion species was $C^{6+}$. In the text, these are labeled as carbon ions for simplicity. On-shot spatially resolved particle detection was conducted with a proton beam profiler. Another calibrated GOS scintillator screen was attached to the rear of the catcher to measure the beam profile of protons with an energy larger than 40 MeV. A bandpass-filtered (540(2) nm) CMOS camera captured the emitted luminescence light.

Electron spectra were recorded shot-to-shot using a magnetic spectrometer positioned in the laser-forward direction, 640 mm from the source. Electrons entered through a 10 mm diameter aperture and were magnetically dispersed and deflected onto a GOS scintillating screen imaged by a CMOS camera. Spectral information was retrieved using a custom code incorporating a General Particle Tracer simulation of the spectrometer response. The system was optimized for electrons with kinetic energies of 1 MeV to 60 MeV and allowed evaluation of electron temperatures in the range of 0.5 MeV to 40 MeV.

All neutron detectors were placed outside the vacuum chamber. Integral neutron production was measured using PND-type Bubble detectors and a Bubble Reader from Bubble Technology Industries, positioned at various locations around the target area. To achieve reasonable bubble counts (~100, to reduce statistical and read-out errors), approximately 10 laser shots were accumulated. The bubble count was then converted into a neutron yield according to the protocol in[64], which considered the detector response from the manufacturer and applied temperature and energy corrections based on the MC-simulated neutron energy spectrum at the bubble detectors' positions.

Traces of the diamond detector ($4 \times 4$ mm$^2$ surface and 500 μm thickness), mounted at 1.5 m from the target, were amplified using a 2 GHz C2-HV Broadband Amplifier by CIVIDEC Instrumentation and recorded with a Tektronix MSO64B oscilloscope (6 GHz bandwidth, 12 bits, 10 GS/s). The main challenge was the prompt detector saturation due to the X-ray photon flash (γ-flash in the following) and the pile-up resulting from harsh radiation and high instantaneous flux, which prevented the measurement of individual fast neutron signals at low time-of-flight. These factors, the high signal originating from the γ-flash and the high interaction rate at the detector caused by the high instantaneous flux, generate the signal undershoot due to the coupling of the amplifier. To minimize these issues, 50 mm of lead filter was added in front of the detector.

Following the X-ray photon flash, the trace exhibited high-frequency noise for approximately 250 ns due to the electromagnetic pulse (EMP) from the laser-plasma interaction. Initially, the EMP noise reached amplitudes up to 200 mV, but it was reduced to a few mV by using EM shielding around the detector and electronics and by using passive low-pass filters in the bias voltage supply cable. The detector was also electrically isolated to decouple its ground from the common ground of the whole accelerator installation. For this purpose, the detector was connected to an independent power supply, and plastic foils were used to prevent metallic contact with other surrounding materials. The oscilloscope was placed outside the laser bunker to prevent ground loops and extra EMP coupling in the acquisition system itself. The saturation due to the γ-flash and the EMP ringing were relatively consistent between shots of a single day; thus, the corresponding exponentially decaying tail and the constant oscillating component were averaged across all recorded traces and subtracted from each trace, resulting in a significantly cleaner baseline that allowed for better signal identification. Signal undershoot and associated baseline shifts were corrected by locally averaging the values immediately before and after the signal.

The detector was energy-calibrated to establish the detection threshold applied in the signal analysis. According to the manufacturer, the energy deposited in the detector for a signal depends directly on the area of the signal and the carbon ionization energy (13 eV), and inversely on the detector impedance (50 Ω), the gain of the amplifier (150), and the electron charge. To avoid integrating residual noise in the signals, the area is calculated assuming the step-function shape of the signals[50], i.e., multiplying the signal's width (FWHM) by its amplitude, since these variables are less affected by noise. To assess the effect of the uncertainties in the signal area and possible gain variation, ±30% fluctuations in the calibration were tolerated, which are within the statistical uncertainties.

### Monte Carlo (MC) simulations

To evaluate the influence of the different secondary particles and to inform the design of the experimental setup, simulations were performed with the MC code PHITS (version 3.31)[83,84], applying input parameters as presented in Fig. 1. The geometry was adapted to the real laboratory infrastructure consisting of concrete walls and floor, vacuum vessel (stainless steel and aluminum assembly) with inner components (such as glass parabola, breadboard, and magnet yokes), and detector system (bubbles, scintillators, and diamond), including applied lead shielding. Both catcher options, LiF and Cu, used in the experiment were evaluated. Within the simulation framework, the following settings were applied: (1) JENDL-4.0[66,85] nuclear data library was used for neutron transport and to model proton-induced nuclear reactions, e.g., (p,n) reactions. (2) TENDL-2019[65] was used for photo-nuclear reactions, and (3) EGS5 (Electron-Gamma Shower, version 5) code[86] for photon and electron transport. (4) The Kurotama[87] and Intra-Nuclear Cascade of Liège (INCL)[88] models were implemented as part of the simulation settings. Simulations of the neutron efficiency of the diamond detector were performed for a simplified geometry with the GEANT4 toolkit[60,89,90] using the QGSP_BIC_AllHP physics list, the JEFF-3.3 nuclear data library[91] and the NRESP model[92] for neutron transport and proton-induced reactions.

### Data availability

The data that support the findings of this study are available on request from the corresponding authors.

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

## Acknowledgments

We acknowledge the outstanding technical and experimental support of the Draco laser group. We also thank Dr. Marcel Reginatto for his insightful guidance and support in the data analysis and interpretation. The authors acknowledge financial support by: - The Spanish Ministry of Economy and Competitiveness (RYC-2014-15271, FPA2016-77689-C2-1-R, RTI2018-098117-B-C21, PID2023-152894OA-I00 and RYC2021-032654I projects); - MAMC has received funding from the EC Euratom research and training program 2014-2018 under grant agreement No 847594 (ARIEL) and the EC PEOPLE Marie-Curie Action program 2007-2013 under grant agreement No 334315 (NeutAndalus); - Xunta de Galicia (grant ED431F2023/21, CIGUS Network of Research Centers); - "La Caixa" Foundation (ID 100010434), fellowship code LCF/BQ/PI20/11760027; - María de Maeztu grant CEX2023-001318-M funded by MICIU/AEI/10.13039/501100011033S; - The project was supported by Hi-Acts, an innovation platform under the grant of the Helmholtz Association HGF and funded by BMFTR under the project PLANET 13N16951- SSch acknowledges support from Trumpf SE + Co. KG; - CR acknowledges support from the LOEWE excellence program of the state of Hesse - This work was supported by CIVIDEC Instrumentation.

## Author contributions

- MAMC, CG, TRG, BF, SSch, JK, CR, AA, JB, RB, AJ, FK, IP, SU, KZ, CW, JMN, MRe, TZ and EG prepared and/or conducted the experiments. - MAMC and SSch performed the simulations. - MAMC, SSch, CG, JK, FK, IP, KZ, JMN, RS, IP, MRe analyzed the data. - CG, MAMC, SSch, FK, KZ, AJ wrote the manuscript. - MAMC, SSch, AA, JB, RB, TEC, BF, EG, AJ, JK, FK, JMN, IP, JMQ, MRe, CR, TRG, US, MRo, RS, SU, CW, KZ, TZ and CG reviewed the manuscript and contributed to discussions. - CG, FK, KZ, AJ, TEC and US supervised the project.

## Funding

## Competing interests

The authors declare no competing interests.
