## [Transparent Peer Review file · Nature Communications]

Single-event fast neutron time-of-flight spectrometry with a petawatt-laser-driven neutron source

Corresponding Author: Dr Carlos Guerrero

Version 0:

Reviewer comments:

Reviewer #1

(Remarks to the Author)

The manuscript reports the development of a laser-driven neutron source (LDNS) and fast neutron energy spectrometer based on a single-event analysis technique. The authors succeeded in generating MeV-energy neutrons over continuous 200 laser shots on the DRACO laser facility. The neutrons ranging from 1 to 30 MeV are analyzed by the single-event spectrometer, capable of resolving individual neutron-induced reactions. The authors also discussed the possibility of laser-driven fast neutron spectroscopy compared with conventional accelerator-based neutron sources. In general, the study of LDNS and its applications is growing rapidly these days. However, in order to attract the interest of many readers from a wide range of fields, the manuscript should be revised, where descriptions that are difficult to understand for researchers outside the field and several inaccurate descriptions need to be changed.

As to the title "Single-event neutron time-of-flight spectroscopy with a petawatt-laser-driven neutron source," the "single event" mode, as the antonym of current mode, is a technical term used in nuclear physics and high-energy physics, and is unfamiliar to researchers in other fields. The authors should reconsider using it in the title. In addition, "Spectroscopy" is generally defined as a fundamental tool to explore the characteristics of matter, including the composition, physical structure, electronic structure and so on. This manuscript only demonstrated the neutron detector system, not showing the result of spectroscopy. In my opinion, "spectroscopy" should be changed into "spectrometer" or "energy analyzer".

In the abstract, the authors described "However, their use in single-event fast neutron spectroscopy remains unproven." This statement is misunderstanding because the demonstration of the single-event spectroscopy by epi-thermal neutrons from LDNS has already been reported in a journal of Nature Springer group [Koizumi et al., Scientific Reports 14, 21916 (2024)], where the areal density of two kinds of metal plates (Ag and In) were successfully analyzed. The authors should refer to the manuscript above and explain the novel point of this manuscript including the difference between fast and epi-thermal neutrons.

On page 3, the authors raised three key capabilities that must be established for the research of neutron-induced nuclear reactions. However, they contain some statements that are not in line with reality. In the item (1), the authors claim that bubble detectors and activation methods are not suitable in the harsh environment of the laser plasma acceleration (LPA). This statement isn't fair. These detection methods provide meaningful results of neutron counting when they're used correctly. Many researchers know from experience that when the bubble detectors are located near the chamber wall (as shown in Fig. 1 of this manuscript), the number of neutrons generated from the LDNS is always overestimated. This is because the neutrons from the chamber wall are directly injected into the bubble detectors. To avoid this, bubble detectors are often placed near the LDNS, where the effect of backgrounds, including neutrons from the chamber wall, is expected to be small. Also in the use of activation method [33], the activation sample was collated directly on the LDNS because of the reason above. The statement by the authors may give an impression that these detectors are always unsuitable in the LPA experiments, but that's not what the authors intended.

In the same item, the authors claim that the current mode operation of neutron scintillator detectors is not suitable for the LDNS experiments. This is misunderstanding, because the current mode operation can serve as a reliable spectrometer when the backgrounds are appropriately subtracted. The current mode detector is historically established by the researchers of nuclear physics [Bowman et al., Nuclear Instruments and Methods in Physics Research Section A: Accelerators, Spectrometers, Detectors and Associated Equipment, 297, 183(1990)], not only by the researchers of LPA, and regarded as

a reliable method. In the field of LPA and LDNS studies, the neutron resonance spectroscopy was successfully obtained by the current mode detector system [21], where the gamma-ray flash was suppressed by the high-speed voltage switch system and the background level was determined from the resonance signal of 100% absorption. (Note the reference 21 is explained as a study of radiographic imaging on Page 2. This is not correct. The radiographic imaging was shown by Yogo et al., Applied Physics Express 14, 106001 (2021)). Recently, the temperature of the samples were analyzed [Lan et al., Nature Communications 15, 5365 (2024)] by the Doppler broadening of the resonance signal obtained in the current mode spectroscopy driven by LDNS. These results should be considered in the manuscript. In addition, the statement on page 5, "In previous LDNS experiments, detection systems have mostly operated in current or integral mode, or relied on passive detectors such as activation foils or bubble dosimeters. While these approaches are suitable for estimating total neutron yields or monitoring source stability, they do not provide the time-resolved and event-specific information required for neutron spectroscopy or reaction studies." should be reconsidered.

In the item (2), the authors claim that "To achieve sufficient statistics, single-event signals need to be accumulated over many laser shots." What is the lower limit of shot number required for the spectroscopy the authors try to demonstrate? 200 shots are sufficient for it? Is the requirement of many laser shots ("many" is not a scientific term) attributed to the neutron number per shot or the limit coming from the pile-up effect of the single-event detector in high neutron flux condition? On page 5, the authors described "Contrary to conventional accelerator-based sources, particles other than protons are also produced in the laser-pitcher interaction and are emitted in different directions. The impact of these is strongly dependent on the specifics of the experimental setup and has been neglected in all previous studies." Please show the evidence that all previous studies neglected the effect.

Fig. 4 (b) shows the comparison between the raw data and the background, where the background value is higher than the raw data for the energy higher than 10 MeV. This result indicates that the statistics is not sufficient in the single laser shot measurement. Is it possible to integrate the data over several laser shots to ensure the statistics? The low statistics results in the large error bars shown in the frame (c) and (d).

Fig. 4(c,d) show the results of MC simulations by Geant 4 as lower limit and PHITS as upper limit. Why does the Geant4/PHITS result show lower/upper limit of the neutron spectrum? Please explain the difference of physics and processes assumed in the two codes.

Figures 4(c) and (d) show that only two or three data points are obtained for each 5-MeV energy range. This indicates that the energy bin is too discrete to analyze the nuclear reactions assumed in this work. Enhancing the statistics will resolve this problem. Why don't the authors integrate the neutron spectra over several laser shots? How many shots are useful for analysis out of 200 laser shots?

On page 9, the authors discuss the energy resolution of their LDNS in the TOF method (see Fig. 5) and compare it with the results of conventional accelerator-based neutron sources.

The DRACO results appear to be simulation-based values (or calculations based on the laser pulse width), not experimental values. If so, the authors should clarify that they are based on simulations to avoid any misunderstanding. In general, comparisons with results from other research fields should be made with caution. If the neutron pulse width in LDNS has not been obtained experimentally, then careful consideration should be given to claiming the superiority of LDNS compared to accelerators. Note that the neutron pulse duration can be evaluated experimentally from the width of the neutron resonance absorption, as described in the manuscripts by Lan and Bowman mentioned above.

In the conclusion, the authors described "An interesting choice of samples and reactions could be fission on ^{235}U and ^{239}Pu in the spirit of the very recent experiment of the NIFFTE Collaboration [65] at LANL for measuring the $^{239}\text{Pu}/^{235}\text{U}$ fission cross section ratio with an unprecedented accuracy of just 1%." However, even after reading the abstract and Introduction of [65], the reviewer cannot understand the importance. The importance may not clear to many readers. Please explain the importance in detail or give another example.

Minor points

Ref [2] is a paper on nuclear fission, not an appropriate example of a paper on neutron application to nuclear physics. The following paper is recommended instead.

"The neutron. its properties and basic interactions" Hartmut Abele
Progress in Particle and Nuclear Physics, 60, 1-81 (2008).
This has over 300 citations.

Ref [1] and [11] are identical.

One of the two can be changed into

"Neutron reactions in astrophysics" R Reifarth, C Lederer and F Käppeler
Journal of Physics G: Nuclear and Particle Physics, 41, 053101 (2014).

Reviewer #2

(Remarks to the Author)

This work by Millan-Callado et. al. presents a significant advance in single event time of flight measurements for laser driven neutron sources. The data presented is convincing that neutron induced reactions are a potential avenue for study at LDNS

with diamond detectors and significant attention to characterizing the difficult radiation environment. I am impressed by the work and recommend it for publication. The raw data, analysis, and text present a strong argument, however I would suggest a few additional details could be added, at the discretion of the authors and editors, to benefit the reader.

1. To measure and distinguish single neutron events from the background is a difficult task. The authors diamond detectors are sensitive to neutrons, charged particles, and high energy photons which are all created in the laser interaction. Experiments with Cu and LiF catchers are fully described with and without the shadow bar. The shadow bar is designed to shield the diamond radiation detector from the direct neutrons and only measure the scattered background, but could itself affect the scattered signal in unanticipated ways. No experiments are described without a catcher target that could assist in determining the neutron background from "other components of the experimental setup and the laser interaction chamber." If these experiments were performed their inclusion would strengthen the result. I understand that beam time is limited.
2. In the same theme, additional information and testing on the shadow bar configuration would be greatly appreciated. The background determination is important enough in the single neutron results presented, 40-50% of the signal, that I believe the shadow bar deserves more attention in the body of the work.
3. Much of the paper describes the Monte Carlo simulations. These simulations take as inputs the interpolated proton, electron and x-ray spectra shown in figure 1 and output the expected contribution to the neutron signal shown in figure 3. Including a few spectra in figure 1 from different shots alongside the interpolations could further convince the reader that these measurements were performed correctly.
4. It is unclear in the figure caption and text if the reported RSD values are for all 204 plotted shots or for only the "unshaded region [which] represents shots with optimized neutron performance (stability within 26 % standard deviation)." I would prefer to have RSD for only the optimized shots and also include the RSD of the optimized shots between different days.
5. In figure 4, the key representation of the single event neutron data, I believe a few extra details would help the reader. In the inset in part a, I suggest that the energy of that single neutron event be included. Where does that single neutron signal fall on the plots in part b and c? And what are the energies of the other two neutron signals at 40 and 90 ns?
6. Finally it is implied that this type of data at DRACO can be collected with ~200 shots per day and that the beam time this paper is based on occurred over several days. Understanding the total number of shots for the entire campaign, the number of shots to produce figure 4 c,d, and a comparison of the background and performance over multiple days would benefit the reader.

Version 1:

Reviewer comments:

Reviewer #1

(Remarks to the Author)

The revisions made by the authors have addressed most of the reviewer's concerns, and the revised manuscript represents a substantial improvement over the previous version. Overall, the reviewer is largely satisfied with the technical clarifications and the refined positioning of the present work.

While the revised manuscript improves the clarity of the scope and novelty of the present work, I note an important remaining issue regarding the treatment of prior studies on laser-driven neutron resonance spectroscopy.

It is appreciated that, through the revision process, the authors have recognized the importance of the work by Koizumi et al., *Sci. Rep.* 14, 21916 (2024) and have modified the abstract to acknowledge prior demonstrations of single-event neutron spectroscopy at lower neutron energies. However, this recognition is currently reflected only by a brief addition in the abstract. Given the direct conceptual relevance of this study to the present work, it should be explicitly cited and discussed in the main text (e.g., in the Introduction and/or Discussion), and included as a formal reference.

In addition, other closely related prior studies on laser-driven neutron resonance absorption spectroscopy are not adequately acknowledged in the main text. In particular, A. Yogo et al., *Phys. Rev. X* 13, 011011 (2023), which was cited as an example of neutron generation in the order of 10^{10} neutrons/shot in the present form, demonstrated single-shot neutron resonance absorption spectroscopy using a laser-driven neutron source, while Z. Lan et al., *Nat. Commun.* 15, 5365 (2024) reported resonance absorption analysis using LDNSs in a current-mode detection scheme. These works are highly relevant for contextualizing the spectroscopic capabilities of LDNSs and for clarifying the distinction between single-event and current-mode approaches, and should therefore be explicitly cited and discussed in the main text.

In its present form, the manuscript acknowledges prior resonance spectroscopy studies only implicitly or partially, which may give readers an incomplete view of the development of the field. I consider it essential that these works be explicitly cited and that their relationship to the present study—particularly in terms of neutron energy regime, detection mode (single-event vs. current-mode), and spectroscopic capability—be clearly discussed in the main text.

I regard this point as mandatory to ensure proper attribution and to accurately position the present work within the existing literature before the manuscript can be accepted.

Reviewer #2

(Remarks to the Author)

I have read the revisions to the paper and am pleased with the improvements. I recommend publication of this final result.

Report from Reviewer 1 on Nature Communications manuscript NCOMMS-25-47297

Reviewer comment #1: The manuscript reports the development of a laser-driven neutron source (LDNS) and fast neutron energy spectrometer based on a single-event analysis technique. The authors succeeded in generating MeV-energy neutrons over continuous 200 laser shots on the DRACO laser facility. The neutrons ranging from 1 to 30 MeV are analyzed by the single-event spectrometer, capable of resolving individual neutron-induced reactions. The authors also discussed the possibility of laser-driven fast neutron spectroscopy compared with conventional accelerator-based neutron sources. In general, the study of LDNS and its applications is growing rapidly these days. However, in order to attract the interest of many readers from a wide range of fields, the manuscript should be revised, where descriptions that are difficult to understand for researchers outside the field and several inaccurate descriptions need to be changed.

Response: We would like to thank the appreciations of the reviewer regarding the interest of this work for a wide range of fields, and the specific comments that have helped us revising the text to make it more precise, and accessible and understandable for the interdisciplinary audience of Nature Communications.

Reviewer comment #2: As to the title “Single-event neutron time-of-flight spectroscopy with a petawatt-laser-driven neutron source,” the “single event” mode, as the antonym of current mode, is a technical term used in nuclear physics and high-energy physics, and is unfamiliar to researchers in other fields. The authors should reconsider using it in the title. In addition, “Spectroscopy” is generally defined as a fundamental tool to explore the characteristics of matter, including the composition, physical structure, electronic structure and so on. This manuscript only demonstrated the neutron detector system, not showing the result of spectroscopy. In my opinion, “spectroscopy” should be changed into “spectrometer” or “energy analyzer”.

Response: We appreciate the reviewer's feedback regarding the accessibility of the title and its precision, particularly for readers outside the nuclear and high-energy physics community. We have considered both suggestions carefully.

We agree that the term "Spectroscopy", despite being technically precise for the measurement technique used in nuclear and particle physics where signals from individual radiation or particle interaction events are recorded and analyzed, rather than averaged over many events; can be widely associated with the application of analyzing matter properties (composition, structure, etc.). While our technique is indeed a foundational step towards such applications, the manuscript primarily focuses on the demonstration and characterization of the measurement methodology itself: the successful detection and analysis of individual neutron events in the challenging petawatt laser environment.

To reflect the scope of the work more accurately, which is the development of the measurement system and method, we have adopted the reviewer's suggestion to use the term "Spectrometry" instead of "Spectroscopy" in the title. We believe this choice reduces ambiguity and aligns better with the technical content of the paper, which establishes a new metrology.

Nevertheless, we respectfully request to retain the term "Single-Event" in the title and text.

In nuclear physics and high-energy physics, "Single-Event" is the precise technical term required to distinguish our method from current-mode or charge-integrated detection methods. Single-Event detection for fast neutrons is the core innovation of our work: moving LDNS measurements from bulk-averaged data to the ability to analyze individual neutron interactions with high fidelity. This mode of operation is critical for studying time-of-flight (TOF) and reaction kinetics under the ultrashort pulse duration characteristic of LDNSs. The term "single-event" directly conveys this fundamental change in measurement capability.

To mitigate the risk of misunderstanding by researchers from other fields, we have ensured that the text provides a clear definition of the concepts "single-event", "single-shot" or "single-particle", with the emphasis in "Single-Event Spectrometry" as a measurement technique where the signals from individual radiation or particle interaction events are recorded and analyzed.

The revised title is now:

Single-event fast neutron time-of-flight spectrometry with a petawatt-laser-driven neutron source

Reviewer comment #3: In the abstract, the authors described "However, their use in single-event fast neutron spectroscopy remains unproven." This statement is misunderstanding because the demonstration of the single-event spectroscopy by epi-thermal neutrons from LDNS has already been reported in a journal of Nature Springer group [Koizumi et al., Scientific Reports¹⁴, 21916 (2024)], where the areal density of two kinds of metal plates (Ag and In) were successfully analyzed. The authors should refer to the manuscript above and explain the novel point of this manuscript including the difference between fast and epi-thermal neutrons.

Response: It is indeed crucial to make the distinction between epithermal and fast neutrons. Furthermore, the work by Koizumi et al. (2024) is very relevant, and in this context is an excellent demonstration of LDNS-based single-event neutron spectroscopy for epithermal neutrons.

Our previous wording, "remains unproven," was indeed too broad and we apologize for the lack of specificity. Although the abstract (and the full manuscript) it was always contextualize in the regime of high-energy neutrons, which is not directly applicable in Koizumi et al. (2024), we have amended the abstract to clearly acknowledge the precedents in other energy regimes and to precisely define the scope and novelty of our work, which lies in the higher energy regime (fast neutrons, $E > 1$ MeV) and the associated technical challenges.

The statement in the Abstract has been modified from:

"However, their use in single-event fast neutron spectroscopy remains unproven..."

to:

"While single-event neutron spectroscopy has been demonstrated with epithermal and low-energy neutrons, its application to fast neutron spectrometry is more challenging and remains unproven."

Clarification of novelty and energy regimes

The critical difference between **epithermal** (or low-energy) and **fast neutrons** defines the novelty of our study:

- **Epithermal/Low-Energy Neutrons (~ 0.5 eV 1 MeV):** These neutrons primarily interact via absorption or scattering at relatively low recoil energies. The work by Koizumi et al. focuses on this regime, which is ideal for material analysis through resonance phenomena and is technically less demanding on the detector's time resolution, signal-to-noise ratio within a harsh environment and fast recovery from the gamma-flash.
- **Fast Neutrons ($E > 1 \text{ MeV}$):** This regime is dominated by inelastic scattering and fast-neutron-induced reactions. Accurate spectroscopy in this regime requires detectors capable of resolving fast recoil protons or alpha particles (e.g., in our diamond detector) operating in the extreme high-energy particle and gamma-ray background generated by the petawatt laser-plasma interaction, which is a significant challenge when the detector must be placed close to the source.

Our main novelty is the successful demonstration of single-event counting spectroscopy for fast neutrons (at only tens of ns after the laser shot) *while operating under the full, unmitigated electromagnetic and radiation background* of the petawatt laser interaction and demonstrating stable multi-shot performance (over 200 shots), which is a key requirement for any future experimental campaign.

Reviewer comment #4: On page 3, the authors raised three key capabilities that must be established for the research of neutron-induced nuclear reactions. However, they contain some statements that are not in line with reality. In the item (1), the authors claim that bubble detectors and activation methods are not suitable in the harsh environment of the laser plasma acceleration (LPA). This statement isn't fair. These detection methods provide meaningful results of neutron counting when they're used correctly. Many researchers know from experience that when the bubble detectors are located near the chamber wall (as shown in Fig. 1 of this manuscript), the number of neutrons generated from the LDNS is always overestimated. This is because the neutrons from the chamber wall are directly injected into the bubble detectors. To avoid this, bubble detectors are often placed near the LDNS, where the effect of backgrounds, including neutrons from the chamber wall, is expected to be small. Also in the use of activation method [33], the activation sample was collated directly on the LDNS because of the reason above. The statement by the authors may give an impression that these detectors are always unsuitable in the LPA experiments, but that's not what the authors intended.

Response: The reviewer is raising a fair point and a potential source of misunderstanding regarding the role and suitability of established detection techniques in the LDNS community. It was not our intention to give the impression that these methods are generally unsuitable in the LPA environment, we fully acknowledge that techniques such as bubble detectors and activation analysis are successfully and routinely employed in LDNS experiments for reliable neutron counting and bulk spectrometry.

The ambiguity stems from the term "single-event" itself. In this context, we are referring to the need of record and analyze each individual particle interaction within the detector to obtain complete information about individual reactions, which is essential for advanced Time-of-Flight (TOF) analysis and specific nuclear reaction cross-section measurements that constitute our proposed future research program. This resolution is not achievable with methods that integrate signals, such as bubble detectors, activation analysis, or current-mode operation.

We have revised item (1) in the text to:

1. Explicitly define the requirement as "single-event resolution".
2. Acknowledge and validate the successful use of passive and current-mode detectors in LDNS experiments for their intended purpose (counting and bulk spectrometry).
3. Refocus the critique by stating that these bulk-integrating methods "are insufficient for this specific requirement" of individual reaction fidelity.

We believe this revision clarifies that our argument is one of functional necessity for a specific advanced measurement technique rather than a general critique of established detectors.

The text is now re-written as follows:

1) It is necessary to obtain as complete information as possible about individual reactions and interaction events (i.e., single-event detection). This high degree of fidelity necessitates a measurement technique that can resolve each individual particle interaction, which is a key requirement for time-of-flight (TOF) analysis and reaction cross-section studies. Consequently, detection methods that integrate over all interactions within their active volume, such as passive neutron detectors (e.g., bubble detectors \cite{Ing1997}), retrospective activation methods \cite{Mori:2021}, or the typical current-mode operation of scintillator detectors \cite{mirfayzi2015calibration} that are commonly and successfully employed in LDNS experiments for neutron counting and bulk spectrometry, are incompatible with this specific requirement. Thus, a single-event fast neutron detection system must be established that is capable of operating in the harsh environment of an LPA and, what is more difficult, very shortly after the laser shot.

Reviewer comment #5: In the same item, the authors claim that the current mode operation of neutron scintillator detectors is not suitable for the LDNS experiments. This is misunderstanding, because the current mode operation can serve as a reliable spectrometer when the backgrounds are appropriately subtracted. The current mode detector is historically established by the researchers of nuclear physics [Bowman et al., Nuclear Instruments and Methods in Physics Research Section A: Accelerators, Spectrometers, Detectors and Associated Equipment, 297, 183(1990)], not only by the researchers of LPA, and regarded as a reliable method. In the field of LPA and LDNS studies, the neutron resonance spectroscopy was successfully obtained by the current mode detector system [21], where the gamma-ray flash was suppressed by the high-speed voltage switch system and the background level was determined from the resonance signal of 100% absorption. (Note the reference 21 is explained as a study of radiographic imaging on Page 2. This is not correct. The radiographic imaging was shown by Yogo et al., Applied Physics Express 14, 106001 (2021)). Recently, the temperature of the samples were analyzed [Lan et al., Nature Communications 15, 5365 (2024)] by the Doppler broadening of the resonance signal obtained in the current mode spectroscopy driven by LDNS. These results should be considered in the manuscript. In addition, the statement on page 5, "In previous LDNS experiments, detection systems have mostly operated in current or integral mode or relied on passive detectors such as activation foils or bubble dosimeters. While these approaches are suitable for estimating total neutron yields or monitoring source stability, they do not provide the time-resolved and event-specific information required for neutron spectroscopy or reaction studies." should be reconsidered.

Response: We appreciate the reviewer pointing out the incorrect description of Reference [21] on Page 2. We have corrected it in the text and we have checked the neighboring references for accuracy.

We also acknowledge the reviewer's detailed point regarding the current-mode operation of scintillator detectors. We agree that current-mode detectors, when properly managed (e.g., background subtraction, gamma flash suppression), have been historically established in nuclear physics and have been successfully employed in LDNS experiments for neutron resonance spectroscopy, but only because this does not require a precise counting of individual neutrons but only the sample-in/sample-out ratio in which the background and efficiency corrections are cancelled, which is not the case for spectrometry or neutron time-of-flight in general, and that is indeed their limitation. However, as it was addressed in the previous comment, our use of the term "unsuitable" was strictly limited to the necessity for single-event resolution in many applications. Both passive detectors and current-mode detectors inherently operate by integrating or averaging the signal over many interactions and over a finite time window. This approach, while excellent for bulk measurements and total yield estimation, does not provide the time-resolved information necessary for the advanced nuclear reaction studies proposed in our manuscript. Our work focuses on establishing this specific single-event fast neutron time-of-flight methodology in the LPA environment.

Considering these discussions, the statement on Page 5 has been revised to remove any suggestion that these methods are generally unsuitable, now focusing solely on the limitation for single-event analysis:

In previous LDNS experiments, detection systems have mostly operated in current or integral mode (relying on a neutron scintillator signal proportional to the total instantaneous rate of interactions) or relied on passive detectors (such as activation foils or bubble dosimeters). While these bulk-integrated approaches may be valid for characterizing the source, providing good estimations of the total neutron yields and establishing the source stability, they fundamentally lack the temporal resolution and event-specific fidelity necessary for time-of-flight measurements and the analysis of individual neutron reaction kinematics as pursued in this work. Furthermore, in LDNS measurements the scintillators are normally surrounded by massive shielding, as in the experiment presented herein, that affect heavily the detector response to both neutron and photons.

Reviewer comment #6: In the item (2), the authors claim that "To achieve sufficient statistics, single-event signals need to be accumulated over many laser shots." What is the lower limit of shot number required for the spectroscopy the authors try to demonstrate? 200 shots are sufficient for it? Is the requirement of many laser shots ("many" is not a scientific term) attributed to the neutron number per shot or the limit coming from the pile-up effect of the single-event detector in high neutron flux condition?

Response: We understand the need of a more precise discussion about the requirements for shot accumulation. Indeed, the term "many" is unscientific and needs clarification, as it touches upon the core challenges of LDNS particle detection experiments. We have modified the text in Item (2) to replace the vague term and explicitly incorporate the inherent variability of this requirement, which is essential context for our work:

(2) To achieve sufficient statistics for reliable neutron time-of-flight analysis in single-event experiments, individual detector signals must be accumulated over multiple laser shots. The precise number of laser shots required is highly variable and experiment-specific, depending on factors such as the detector's intrinsic efficiency, the solid angle coverage, the target nuclear cross-section, and the desired energy resolution. For instance, ~200 laser shots are sufficient for the demonstration experiment presented herein, but more than 10^5 proton beam pulses at the CERN n_TOF spallation source have been necessary recently to measure accurately the $^{12}\text{C}(n,p/d)$ reactions up to 25 MeV [Zugec:2025]. This inherent requirement for accumulation dictates the need for stable, high-repetition-rate source operation and shot-to-shot monitoring of the experimental conditions to ensure data integrity.

As noted in the revised text, the required number of shots varies and is determined by a complex interplay of factors:

- Necessity for accumulation (neutron yield/efficiency limit): the primary driver for accumulating hundreds or thousands of shots is the need to populate small energy bins for spectrum analysis. This requirement stems from the low intrinsic detection efficiency (a necessary design choice to mitigate pile-up) combined with the need to measure specific nuclear cross-sections, especially those that are small or in the high-energy tail.
- Sufficient statistics for this work: the 200 shots delivered in the DRACO experiment were sufficient for the goal of this manuscript: to successfully demonstrate the feasibility of the single-event TOF metrology in the harsh PW environment and to benchmark the measured reaction rates against MC simulations. Full, high-resolution spectroscopy would indeed demand a higher number of accumulated events, highlighting the importance of the stable, multi-shot platform we have established.

We believe the revised statement provides the necessary context and scientific justification for the requirement of sustained, high-repetition-rate operation.

Reviewer comment #7: On page 5, the authors described “Contrary to conventional accelerator-based sources, particles other than protons are also produced in the laser-pitcher interaction and are emitted in different directions. The impact of these is strongly dependent on the specifics of the experimental setup and has been neglected in all previous studies.” Please show the evidence that all previous studies neglected the effect.

Response: The use of the term "all" is inappropriate in scientific literature as it is difficult to prove a negative universal. However, we have not found a single reference in which such secondary neutrons sources have been considered. In order to keep our claim of being the first to study this issue in detail while avoiding claiming a complete absence of consideration, we have changed the text to:

"... and, to the best of the author's knowledge, has not been addressed in current LDNS literature..."

Reviewer comment #8: Fig. 4 (b) shows the comparison between the raw data and the background, where the background value is higher than the raw data for the energy higher than 10 MeV. This result indicates that the statistics is not sufficient in the single laser shot

measurement. Is it possible to integrate the data over several laser shots to ensure the statistics? The low statistics results in the large error bars shown in the frame (c) and (d).

Response: This observation provides a valuable opportunity to clarify the nature of single-event spectrometry and the data acquisition presented in Figure 4.

The reviewer's premise that Figure 4 (b) is a single-laser-shot measurement is incorrect. Figure 4 (b), (c), and (d) already represent the cumulative results of ~200 laser shots. We have clarified this fact in the main text: *"To determine the neutron spectrum generated at DRACO~PW, signals from individual neutron interactions were measured using the diamond detector and accumulated over hundreds of laser shots for two catcher materials: LiF (160 laser shots, with ~ 2.3 signals/shot) and Cu (193 laser shot, with ~ 1.2 signals/shot)."* and have explicitly updated the caption for Figure 4 to state *"Neutron energy histograms accumulated over 67 laser shot with (background) and 193 laser shots without (raw) the shadow-bar} for p-Cu pitcher-catcher configuration showing the effect of the background "*

We agree that the low statistics in the high-energy region leads to large error bars in Figures 4 (c) and (d). This is an inherent limitation of this initial proof-of-concept experiment. Our primary goal was to demonstrate the feasibility of the single-event methodology in LDNS—achieving stable operation, managing the extreme background environment, and extracting any measurable reaction signal—, not to deliver high-precision time-of-flight data. The ~200 accumulated shots were sufficient to demonstrate the single-event capability and to validate the resulting reaction rates against Monte Carlo simulations.

Improving the statistics in the high-energy tail will be the objective of future campaigns by using a LPA set-up that can deal with higher repetition rate and total number of laser shots, thus increasing the total number of shots accumulated (as suggested in our discussion), and employing detector arrays.

To deal with the low statistics and optimize the signal-to-background ratio in the ToF spectra, we adjusted the energy resolution (binning) to 2 MeV.

Reviewer comment #9: Fig. 4(c,d) show the results of MC simulations by Geant 4 as lower limit and PHITS as upper limit. Why does the Geant4/PHITS result show lower/upper limit of the neutron spectrum? Please explain the deference of physics and processes assumed in the two codes.

Response: Comparing results from different Monte Carlo codes is a common and essential practice in nuclear and high-energy physics. This is done to establish the model-dependent uncertainty inherent in simulating complex particle interactions, such as those governing neutron production and transport in a pitcher-catcher setup. Discrepancies between codes, particularly in the prediction of differential cross-sections for hadron-nucleus interactions, are well-known. In this way, the band defined by these two established codes provides an estimate (actually, a lower limit)of the systematic uncertainty of the expected neutron yield against which the measurements is compared.

We have modified the text near Figure 4 to briefly include this explanation, ensuring the reader understands the physical origin of the uncertainty band:

Furthermore, these experimental results for both catcher materials (LiF and Cu) were subjected to a validation against Monte Carlo (MC) simulation predictions from Geant4 and PHITS, using in both cases nuclear data from the JENDL-4 evaluated library. The uncertainty band between the codes arises from differences in the particle transport methods, nuclear interaction models and cross-section data employed by Geant4 and PHITS. These discrepancies between different codes in modeling high-energy particle-induced reactions and subsequent neutron transport are well-known in the nuclear physics community (see for instance <https://doi.org/10.1140/epja/i2016-16100-8>). The band shown in Fig.~\ref{Fig_Diamond}c and d therefore serves as an estimate (with its range defining a lower-limit for the associated uncertainty) of the neutron spectrum, against which the experiment is compared.

Reviewer comment #10: Figures 4(c) and (d) show that only two or three data points are obtained for each 5-MeV energy range. This indicates that the energy bin is too discrete to analyze the nuclear reactions assumed in this work. Enhancing the statistics will resolve this problem. Why don't the authors integrate the neutron spectra over several laser shots? How many shots are useful for analysis out of 200 laser shots?

Response: Indeed, the data presented in Figures 4(b), (c), and (d) already represent the cumulative results of hundreds of laser shots. This is very important and, thanks also to a previous comment, it is now addressed more clearly in the text, which now includes the precise number of laser shots (160 for LiF, 193 for Cu and 67 for shadow-cone) as well the average number of detector signals per laser-shot

The statistics achieved with the accumulation of ~200 shots is limited, however that was the maximum number of valid shots that could be delivered within a single day at DRACO, and it's outstanding for this kind of facilities.

The available statistics sets the limit for the neutron energy bin width: if they are narrower, the error bars become too large, thus we believe a that a 2 MeV binning is reasonable for the purpose of the experiment: the feasibility of the single-event TOF method in the harsh PW environment. Although quite broad, it allows a fare comparison with the expected neutron spectrum, which is smooth and thus does not require a fine binning. Being this said, future measurements should profit from the continuous increase in shot frequencies that new developments are allowing, thus providing the means to achieve higher statistics, reducing the energy bin width, and opening the door to study narrow structures such as neutron resonances.

Reviewer comment #11: On page 9, the authors discuss the energy resolution of their LDNS in the TOF method (see Fig. 5) and compare it with the results of conventional accelerator-based neutron sources.

The DRACO results appear to be simulation-based values (or calculations based on the laser pulse width), not experimental values. If so, the authors should clarify that they are based on simulations to avoid any misunderstanding. In general, comparisons with results from other research fields should be made with caution. If the neutron pulse width in LDNS has not been obtained experimentally, then careful consideration should be given to claiming the superiority of LDNS compared to accelerators. Note that the neutron pulse duration can be evaluated

experimentally from the width of the neutron resonance absorption, as described in the manuscripts by Lan and Bowman mentioned above.

Response: The referee is right recommending caution when comparing facilities. Indeed, we believe that a few, but important, details were missing in the text so that any reader can understand and judge what is being compared. In the revised text it is explicitly mentioned that the energy resolution values considered for DRACO are based on Geant4 Monte Carlo simulations starting from a conventional “ps” proton pulse (Dromey et al., 2026), that indeed has not been measured in this experiment. Now we also mention that to fully exploit the excellent time resolution of LDNS the set-up must be combined with high performance fast detectors, of which a few examples are referenced.

Reviewer comment #12: In the conclusion, the authors described “An interesting choice of samples and reactions could be fission on ^{235}U and ^{239}Pu in the spirit of the very recent experiment of the NIFFTE Collaboration [65] at LANL for measuring the $^{239}\text{Pu}/^{235}\text{U}$ fission cross section ratio with an unprecedented accuracy of just 1%.” However, even after reading the abstract and Introduction of [65], the reviewer cannot understand the importance. The importance may not clear to many readers. Please explain the importance in detail or give another example.

Response: After having demonstrated the feasibility of doing using LDNS for ToF experiments in nuclear physics, we believe that it is important for the reader to visualize how an actual cross section measurements could be carried out, that’s why values of pulse frequency, detector configuration and time duration are given for an experiment that could be comparable to the state of the art in the field, illustrated by the mentioned work by the NIFFTE Collaboration. That experiment represents the state-of-the art in neutron reaction studies in terms of accuracy, wealth of observables and impact of the results.

We hope that the inclusion of two sentences in this last paragraph clarifies the situation.

Reviewer comment #13:

Minor Points:

- Ref [2] is a paper on nuclear fission, not an appropriate example of a paper on neutron application to nuclear physics. The following paper is recommended instead: Hartmut Abele, "The neutron. its properties and basic interactions" , Progress in Particle and Nuclear Physics, 60, 1-81 (2008). This has over 300 citations.
- Ref [1] and [11] are identical. One of the two can be changed into "Neutron reactions in astrophysics" R Reifarh, C Lederer and F Käppeler, Journal of Physics G: Nuclear and Particle Physics, 41, 053101 (2014).

Response: We thank the reviewer for these precise and helpful suggestions regarding the references. All corresponding citations in the text have been updated to reflect these changes.

Report from Reviewer 2 on Nature Communications manuscript NCOMMS-25-47297

General comment: This work by Millan-Callado et. al. presents a significant advance in single event time of flight measurements for laser driven neutron sources. The data presented is convincing that neutron induced reactions are a potential avenue for study at LDNS with diamond detectors and significant attention to characterizing the difficult radiation environment. I am impressed by the work and recommend it for publication. The raw data, analysis, and text present a strong argument, however I would suggest a few additional details could be added, at the discretion of the authors and editors, to benefit the reader.

Response:

We would like to thank the reviewer for the positive assessment of our work and the recommendation for publication. The revision based on the helpful comments has helped us to improve the manuscript.

Reviewer comment #1: To measure and distinguish single neutron events from the background is a difficult task. The authors diamond detectors are sensitive to neutrons, charged particles, and high energy photons which are all created in the laser interaction. Experiments with Cu and LiF catchers are fully described with and without the shadow bar. The shadow bar is designed to shield the diamond radiation detector from the direct neutrons and only measure the scattered background but could itself affect the scattered signal in unanticipated ways. No experiments are described without a catcher target that could assist in determining the neutron background from "other components of the experimental setup and the laser interaction chamber." If these experiments were performed their inclusion would strengthen the result. I understand that beam time is limited.

Response: Indeed, time at a PW laser facility is very limited, but we attempted the measurement indicated by the reviewer by means of a polyethylene catcher that could have served as dummy. However, the maximum energy of the LPA protons was very high and produced neutrons when reacting with carbon nuclei. The text now includes this information:

"Ideally, an additional measurement without a catcher would help understanding the nature of the background unrelated to neutrons. This was attempted using a polyethylene catcher as a dummy, but with proton beam energies exceeding 20~MeV even the $^{12}\text{C}(p,n)$ reaction channel ($Q=-18$ MeV) was open, and the measurement produced a measurable neutron field (see shot ~ 30 in Fig. 3)."

Reviewer comment #2: In the same theme, additional information and testing on the shadow bar configuration would be greatly appreciated. The background determination is important enough in the single neutron results presented, 40-50% of the signal, that I believe the shadow bar deserves more attention in the body of the work.

Response: The background is indeed comparable to the direct neutron components and the shadow bar measurement, as well as the simulations, was key to assess it. The technique is widely used for this purpose.

In the revised text, we have added to the text our design calculations and estimates on the efficiency of the shadow bar:

“This shadow bar, although limited in size because of the lack of space, acted as an efficient neutron shield \cite{shadowbars}: Geant4 Monte Carlo simulation showed that it absorbed or scattered 99,9\% of the incident neutrons, and as much as 99,0\% if only the more penetrating high energies (>10~MeV) are considered.”

Reviewer comment #3: Much of the paper describes the Monte Carlo simulations. These simulations take as inputs the interpolated proton, electron and x-ray spectra shown in figure 1 and output the expected contribution to the neutron signal shown in figure 3. Including a few spectra in figure 1 from different shots alongside the interpolations could further convince the reader that these measurements were performed correctly.

Response: While reasonable in other contexts, the request by the reviewer to show more spectra of different shots in Fig-.1 is not applicable in this case. Taken as example the proton spectra, which is the most important input, it has a complex structure (broad spectrum and strong energy dependent divergence) that cannot be monitored without affecting the beam itself. Therefore, we have used a couple of detector systems based on different principles, each providing a different piece of information the is then combined to estimate the proton beam intensity and energy distribution. This is sketched in fig 1b and key metrics are then collected in fig3. Now overlaying e.g. TPS spectra, which would need to be scaled to full solid angle, and which are only valid (absolutely calibrated against Lanex) in certain energy ranges and are complemented below and above such angle with data from other systems and previous measurements, would be incomplete and misleading. The same applies to x-rays and electrons.

We believe that there is enough literature about LPA experiments at DRACO and also about the techniques to characterize the proton, x-rays and electrons fields produced, and therefore believe that it is better to not add further spectra to the manuscript.

Reviewer comment #4: In it unclear in the figure caption and text if the reported RSD values are for all 204 plotted shots or for only the "unshaded region [which] represents shots with optimized neutron performance (stability within 26 % standard deviation)." I would prefer to have RSD for only the optimized shots and also include the RSD of the optimized shots between different days.

Response: It is indeed unclear. We have now clarified in both the figure caption and the main text that the RSDs given correspond to the consecutive 193 laser shots on the Cu target.

Reviewer comment #5: In figure 4, the key representation of the single event neutron data, I believe a few extra details would help the reader. In the inset in part a, I suggest that the energy

of that single neutron event be included. Where does that single neutron signal fall on the plots in part b and c? And what are the energies of the other two neutron signals at 40 and 90 ns?

Response: It is true that it is difficult to correlate the time-of-flight in 4 a) with the neutron energies displayed in b), c) and d), as the relationship is not linear. Now Fig. 4a features a second X-axis in the top showing the neutron energy, which we think it is indeed very useful for the reader.

Reviewer comment #6: Finally, it is implied that this type of data at DRACO can be collected with ~200 shots per day and that the beam time this paper is based on occurred over several days. Understanding the total number of shots for the entire campaign, the number of shots to produce figure 4 c,d, and a comparison of the background and performance over multiple days would benefit the reader.

Response: This is partially explained in the methods section. The Plasma Mirror (PM) operation limited the shot rate to about one shot every 20 seconds and the total number of shots to about 200 shots per day, but now the description in the spirit of the reviewer's comment:

"PM operation limited the shot rate to about one shot every 20 seconds and the total number of shots to about 200 shots per day. The work presented herein correspond to neutrons generated using two different catcher options ($\sqrt{10}$ mm thick LiF and $\sqrt{3}$ mm thick Cu) on two separate days. The complete experimental campaign took place over several days, which were devoted to the optimization of the LPA and detection systems"

2nd report from Reviewer 1 on Nature Communications manuscript NCOMMS-25-47297

The revisions made by the authors have addressed most of the reviewer's concerns, and the revised manuscript represents a substantial improvement over the previous version. Overall, the reviewer is largely satisfied with the technical clarifications and the refined positioning of the present work.

While the revised manuscript improves the clarity of the scope and novelty of the present work, I note an important remaining issue regarding the treatment of prior studies on laser-driven neutron resonance spectroscopy.

It is appreciated that, through the revision process, the authors have recognized the importance of the work by Koizumi et al., *Sci. Rep.* 14, 21916 (2024) and have modified the abstract to acknowledge prior demonstrations of single-event neutron spectroscopy at lower neutron energies. However, this recognition is currently reflected only by a brief addition in the abstract. Given the direct conceptual relevance of this study to the present work, it should be explicitly cited and discussed in the main text (e.g., in the Introduction and/or Discussion), and included as a formal reference.

In addition, other closely related prior studies on laser-driven neutron resonance absorption spectroscopy are not adequately acknowledged in the main text. In particular, A. Yogo et al., *Phys. Rev. X* 13, 011011 (2023), which was cited as an example of neutron generation in the order of 10^{10} neutrons/shot in the present form, demonstrated single-shot neutron resonance absorption spectroscopy using a laser-driven neutron source, while Z. Lan et al., *Nat. Commun.* 15, 5365 (2024) reported resonance absorption analysis using LDNSs in a current-mode detection scheme. These works are highly relevant for contextualizing the spectroscopic capabilities of LDNSs and for clarifying the distinction between single-event and current-mode approaches and should therefore be explicitly cited and discussed in the main text.

In its present form, the manuscript acknowledges prior resonance spectroscopy studies only implicitly or partially, which may give readers an incomplete view of the development of the field. I consider it essential that these works be explicitly cited and that their relationship to the present study—particularly in terms of neutron energy regime, detection mode (single-event vs. current-mode), and spectroscopic capability—be clearly discussed in the main text.

I regard this point as mandatory to ensure proper attribution and to accurately position the present work within the existing literature before the manuscript can be accepted.

Response:

We would like to thank the reviewer for the positive assessment of our work and the recommendation for publication.

We understand that the suggestions regarding a more detailed reference to previous works on neutron resonance spectroscopy are reasonable and have thus added the necessary text and references in the main text of the paper. The new text and references are marked in red for easier revision and are also given in the following:

LDNSs have been experimentally realized and investigated, most commonly by directing LPA protons or deuterons onto neutron converter targets in various configurations \cite{roth2013bright, Lelievre2023, Jiao2023} (see Ref. \cite{Mirfayzi:2025} for the most recent review). Proof-of-concept experiments have shown the use of laser-driven neutrons for radiographic imaging \cite{Yogo2021}, material analysis \cite{Zimmer2022, Mirani2023}, and as diagnostic tools in laser-plasma physics \cite{Alejo2017, Alejo2022, Yao2023}. Among these, the most studied application compatible with current LDNS capabilities is neutron resonance spectroscopy. This is a non-destructive material analysis technique where the dips in a transmission measurement are unambiguously assigned to a given isotope. The technique has been achieved both in a single shot, operating a neutron detector in current mode \cite{Lan2024}, and by averaging over a few shots, employing single-neutron detection \cite{Yogo2023, Koizumi2024}. However, to date, experimental work has been limited to moderated sources delivering epithermal neutrons in the eV range. For unmoderated fast-neutron sources, only prospective studies on resonance spectroscopy or single-neutron detection have been reported \cite{Mirani2023}. Consequently, more complex measurements approaching full nuclear-physics experiments remain unexplored. Even if they have been recognized as a key scientific case at upcoming world-class laser facilities like ELI-NP \cite{ELI-NP:2016}, their feasibility has yet to be experimentally demonstrated.